# Weakly Supervised Virus Capsid Detection with Image-Level Annotations in Electron Microscopy Images

**Hannah Kniesel,**\* **Leon Sick, Tristan Payer, Tim Bergner, Kavitha Shaga Devan, Clarissa Read, Paul Walther, Timo Ropinski**
Ulm University

**Pedro Hermosilla**
TU Vienna

## Abstract

Current state-of-the-art methods for object detection rely on annotated bounding boxes of large data sets for training. However, obtaining such annotations is expensive and can require up to hundreds of hours of manual labor. This poses a challenge, especially since such annotations can only be provided by experts, as they require knowledge about the scientific domain. To tackle this challenge, we propose a domain-specific weakly supervised object detection algorithm that only relies on image-level annotations, which are significantly easier to acquire. Our method distills the knowledge of a pre-trained model, on the task of predicting the presence or absence of a virus in an image, to obtain a set of pseudo-labels that can be used to later train a state-of-the-art object detection model. To do so, we use an optimization approach with a shrinking receptive field to extract virus particles directly without specific network architectures. Through a set of extensive studies, we show how the proposed pseudo-labels are easier to obtain, and, more importantly, are able to outperform other existing weak labeling methods, and even ground truth labels, in cases where the time to obtain the annotation is limited.

## 1 Introduction

Deep learning algorithms rely on large data sets for training a model to perform a complex task. However, annotating such large data sets usually requires a person to analyze each data point and label it accordingly, resulting in a time-consuming process. In particular, detecting particles in Electron Microscopy (EM) images is extremely costly, since this annotation process has to be performed by an expert, which usually results in small data sets that are not well suited to train deep models. Since particle detection is a key step in several scientific studies such as the analysis of the formation of infectious virions (Shaga Devan et al., 2021), catalytic investigation (Nartova et al., 2022), analysis of multi-tissue histology images (Graham et al., 2019), preclinical trials or single particle reconstruction (Sigworth, 2015; Shaikh et al., 2008), these studies could benefit significantly from automated object detection methods tailored towards particle detection. However, particle detection comes with additional challenges. First, it requires the need for experts to annotate the data. Second, since these areas are active research fields, they require a quick adaption of the detection model to new virus mutants, particles, or imaging modalities.

To address this issue, weakly supervised algorithms (Oquab et al., 2015; Bency et al., 2016; Zeng et al., 2019; Gao et al., 2021) rely on a secondary task, usually classification, for which annotations are easy to obtain. Then, to solve the main task of object detection, weakly supervised algorithms usually use a set of bounding box candidates or Region of Interests (ROIs), obtained with a selective search strategy (Uijlings et al., 2013), which are later filtered based on the classification score of a pre-trained classification model. However, the accuracy of such object detection models highly depends on the quality and quantity of such ROI candidates (Girshick, 2015), since a direct regression of bounding

---

\*Corresponding author email: hannah.kniesel@uni-ulm.de

boxes based on the pre-trained classifier is not possible. This is why current weakly supervised methods in the field of particle detection in EM usually rely on more fine-grained (and expensive) object-level annotations rather than image-level annotations (Devan et al., 2019; Matuszewski & Sintorn, 2019).

In our work, however, we reduce annotation time by exploiting image-level annotations for virus capsid detection in EM images, by proposing a distillation method, that is able to regress the bounding box position directly from a classifier pre-trained on image-level annotations. To this end, we combine a Gaussian masking strategy and domain-specific knowledge about the virus size and its shape, in order to localize virus capsids on the images using an optimization algorithm informed by the pre-trained classifier. To propagate the gradients over the full input image, we initialize the Gaussian mask with a large standard deviation and progressively reduce it during the optimization procedure, similar to the training mechanism used in score-based generative models (Song & Ermon, 2019). By exploiting this novel approach, we are able to perform accurate particle detection, which is robust with respect to the variance of the initial ROIs. Since our approach is only relying on image level labels, the collection of a new data set for a newly discovered virus mutant or a new imaging modality can be done efficiently. To evaluate our methods, we first conducted a user study comparing different types of labels which results show that our labels are easier to obtain and less prone to errors. Then, we compare our approach to other weakly supervised and fully supervised approaches on five different virus types. Our results show that our approach, solely relying on image labels, does not only outperform other weakly supervised approaches but even fully supervised ones when allocating the same annotation time. Thus, within this paper, we make the following contributions:

- We propose a domain-specific gradient-based optimization algorithm, which exploits a pre-trained classifier and a Gaussian masking strategy, in order to detect virus capsids.

- We introduce a class activation map guided initialization strategy to significantly reduce the computational overhead of the underlying optimization process.

- We conducted a user study comparing different label types and show that image-level annotations are easier and faster to obtain, and more robust to annotation errors.

- We show that our approach outperforms other weakly as well as fully supervised methods given the same annotation time.

## 2 RELATED WORK

**Weakly supervised object detection.** The requirement for fast annotation times is a long-standing problem in several fields, which has made Weakly Supervised Object Localization (WSOL) and Weakly Supervised Object Detection (WSOD) an active area of research in the last few years.Oquab et al. (2015) introduced a CNN architecture that can be moved over the input image during inference time in a sliding window fashion to perform WSOD. Bazzani et al. (2016) used a selective search strategy (Uijlings et al., 2013) to draw a set of bounding box candidates on the image for which the score of each box was obtained from a pre-trained classification model. Bency et al. (2016) used a hierarchical search to reduce the number of bounding box candidates and the feature map of a deep network to find the location of the object of interest. Bilen & Vedaldi (2016) introduced Multiple Instance Learning (MIL) in an end-to-end trainable fashion. In MIL, training instances are organized in bags such that a positive bag contains at least one object of interest and a negative bag does not contain any object of interest. There are many works to follow and improve upon the MIL approach (Kantorov et al., 2016; Diba et al., 2017; Tang et al., 2017; Cheng et al., 2020; Huang et al., 2020; Ren et al., 2020; Zeng et al., 2019; Seo et al., 2022). Among these methods, the approach by Zeng et al. (2019) stands out, as it includes refinement of the ROI candidates in the loss to obtain more accurate bounding box predictions. A similar idea for bounding box refinement was also explored by Dong et al. (2021). However, they rely on additional data sets to learn bounding box modifiers that can be applied to the data set with weak labels. Contrary to these approaches, we propose to directly regress the bounding box of the objects from the pre-trained classifier without the need for supervised pre-training on different data sets, while further being robust to initial ROI proposals computed by selective search (Uijlings et al., 2013) or similar methods. This makes it possible to use a smaller amount of initial ROI candidates to reduce computational cost.

In another line of work, researchers have investigated how to obtain the object's bounding box from Class Activation Maps (CAMs) of pre-trained deep neural networks (Zhou et al., 2015). However, such methods have difficulties identifying specific discriminative parts of objects. To address this problem, Singh & Lee (2017) tried to improve the quality of CAMs by randomly masking patches of the image during the training phase, to not only rely on specific features of the object during predictions. This concept was later extended by Zhang et al. (2018) and Choe & Shim (2019), whereby both methods facilitate attention maps to mask certain regions of the image during training. Later, Xue et al. (2019) proposed a regularization loss and a hierarchical classification loss to enforce discrepancy in the feature maps, which allows the classifier to attend to the full extent of objects. More recently, Gao et al. (2021) investigate the attention mechanism of vision transformers (Dosovitskiy et al., 2020) to guide WSOD. Alternatively, Meng et al. (2021) aim to better capture the full object through object-aware and part-aware attention. With a similar goal, Wei et al. (2022) propose a mechanism that enforces inter-class feature similarity and intra-class appearance consistency, while Xu et al. (2022) use class-specific foreground and background context embeddings which act as class centroids during inference to form a more complete activation map. However, those methods still suffer from correctly identifying the full extent of objects and/or rely on specific architectures which might not be suited for small data sets, as they usually occur in EM scenarios.

The most similar work to the one proposed in this paper is the work from Lu et al. (2020). They propose a secondary network to predict the geometric parameters of a mask, center and radius of an ellipsoid, which is then input to another network that predicts the final mask. They show in their experiments that using the predicted geometry directly leads to poor performance. In this work instead, we show that no neural network is necessary to predict or transform such mask and by using similar ideas to the ones used in score-based generative models (Song & Ermon, 2019) we can optimize directly the location of the object. Additionally, we introduce a method that is able to detect multiple instances of the same object in an image, which is not possible with the approach of Lu et al. (2020).

**Virus particle detection in EM.** Despite the progress in WSOD in standard computer vision, its application in EM images is limited, even though the need for fast annotations in EM is of special interest. This is likely the case since low Signal to Noise Ratio (SNR) in EM images can limit the capacity of methods well performing on other imaging modalities. Further, EM data usually contains a high number of instances of the same object in one image, which are hard to detect in a weakly supervised setup with image-level labels only. To solve a similar problem in medical imaging, Dubost et al. (2020) introduced regression models for WSOD in 3D MRI data. In the EM domain Huang et al. (2022) introduced a weakly supervised learning schema for finding the location of proteins in cryo-EM. However, in their weakly supervised setup, they still require a small amount of labeled training data. For detecting virus particles in EM in a weakly supervised fashion Devan et al. (2019) trained a classification model on a small set of annotated bounding box crops of the HCMV nucleocapsids. They then use a weakly supervised approach similar to Oquab et al. (2015) to detect virus particles based on their classifier. However, this approach requires images of a single virus, instead of random crops with and without virus particles. The same authors later explore the improvement of supervised virus detection by augmenting training data by a generative adversarial network (Shaga Devan et al., 2021). One of the most promising works in weakly supervised detection and segmentation probably originates from Matuszewski & Sintorn (2018). They introduced a minimal annotation strategy for the segmentation of microscopy images: annotations of the center or center line of a target object are used to generate segmentation masks. The labels for the object of interest were generated by dilating each particle annotation with a disk of $0.7\times$ average known size of the target object. The background label, on the other hand, was created by dilating the center annotations with $1.5\times$ the average known size of the target object. Later, the same authors (Matuszewski & Sintorn, 2019) made use of the minimal labels to train an improved U-Net architecture (Ronneberger et al., 2015) for virus recognition in EM. However, all of the mentioned methods rely on more fine-grained annotations and/or the use of a compute inefficient sliding window to obtain ROI candidates to locate the particles.

## 3 METHOD

Our method expects as input an EM image, $I \in \mathbb{R}^{W \times H}$, the expected virus radius $r$, and a classifier $C : \mathbb{R}^{W \times H} \to \mathbb{R}$. We pre-train the classifier on image-level annotations, such that it can classify

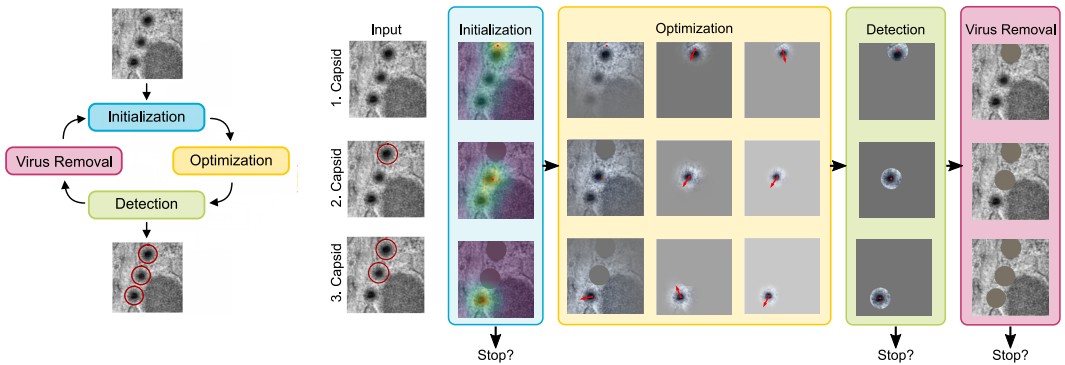

Figure 1: Left: Overview of our weakly supervised virus detection approach, working in an iterative fashion until a stopping criteria is met. Right: Detailed description of our approach visualizing different steps for the detection of a virus. For the *Initialization* of the particle position $p_0$ we compute a CAM obtained through GradCAM (Selvaraju et al. (2017)) and place $p_0$ at the position of the highest CAM value. During *Optimization* the position $p_t$ is iteratively refined, guided by the classifier output and a Gaussian mask with decreasing standard deviation centered at $p_t$. A *Detection* is happening once the position is converged to the exact position of the virus particle. Finally, the input image is prepared to detect the next virus by the *Virus Removal* of previously detected virus particles. We check at multiple points of the virus detection pipeline, if a stopping criteria is met. For more details see section 3.4.

$I$, based on the presence and absence of virus capsids in $I$, in a binary manner. To locate the virus capsids, we first initialize their position $p_0$ with the location of the highest value of a CAM obtained for $C(I)$. This position is then iteratively optimized to obtain a refined position $p_t$ over time steps $t$. In each step $t$, we mask the input image with a Gaussian mask $M$ centered at $p_t$, before optimizing $p_t$ to maximize the classifier score $C(I \cdot M(p_t))$. During this optimization, we fix the weights of the classifier and only optimize the position. In order to successfully converge to the desired position, even when $p_0$ is far from a virus particle, the gradient computation needs to consider areas of $I$ far from $p_0$. Therefore, the standard deviation of the Gaussian is chosen to initially span the entire image and continuously decreases during the optimization process. Once the optimization process converges, the already detected virus particles are cut out from $I$, using virus radius $r$, such that the optimization towards a new viral particle is not misguided by already located particles. This iterative process stops when $C(I)$ predicts the absence of viruses on the masked image. See Figure 1 for an overview of our method.

## 3.1 INITIALIZATION

We compute the CAM of the input image $I$ using the GradCAM (Selvaraju et al., 2017) algorithm based on our pre-trained classifier (implementation from Gildenblat & contributors (2021)). Then, the center of mass of the top 1% of activations is used as the initial position, $p_0$. Thus, if there are multiple instances of the virus, the CAM can spread over multiple regions of the input image. Therefore, to ensure we initialize $p_0$ inside of a relevant region, we check if the center of mass lies within the top 1% of the activations. If this is not the case, we define $p_0$ as a random position among those top 1%.

## 3.2 OPTIMIZATION

Given an initial position $p_0$, we want to further optimize it using gradient descent to match the exact location of a virus. To achieve this, we define a fully differentiable mask $M \in \mathbb{R}^{W \times H}$ as a Gaussian function centered at $p_t$, where $t$ it the current optimization iteration. This mask $M$ is defined as:

$$M_{ij}(p_t) = \frac{1}{\sigma_t \sqrt{2\pi}} \exp\left(\frac{-\|x_{ij} - p_t\|^2}{2\sigma_t^2}\right) \tag{1}$$

where $x_{ij}$ is the coordinate of the position $i, j$ in the image, and $\sigma_t$ is the mask's standard deviation at $t$. Note, that the mask is normalized to have an integral equal to one since we found this to work

$\sigma_{\max}$ $\qquad\qquad\qquad\qquad\qquad\qquad\qquad\qquad\qquad\qquad\qquad\qquad$ $\sigma_{\min}$

Figure 2: Visualization of the magnitude and direction of the gradients for multiple positions in the input image over the optimization process. For large values of the standard deviation, $\sigma_{\max}$, large portions of the image receive gradients pointing to virus particles. However, for small values of the standard deviation, $\sigma_{\min}$, only the regions close to virus particles contain strong gradients pointing towards particles. By reducing the value of $\sigma$ during the optimization process we are able to accurately find a particle in the image even if the initial starting position is far away from any virus.

better in practice. Also, before we feed the masked input to the classifier, we normalize it based on the statistics of the pre-training data set. Then, the optimization objective is defined as:

$$\max_{p_t} \; C(I \cdot M(p_t)) \tag{2}$$

**Mask standard deviation.** In order to propagate gradients to optimize the position over the full EM image, the standard deviation of the Gaussian mask needs to be adapted for each optimization step. While a Gaussian mask with a large standard deviation $\sigma_{\max}$ pulls positions that are far from a virus closer to the optimal position, a Gaussian mask with a small standard deviation $\sigma_{\min}$ will generate smooth gradients for positions close to a virus (see Figure 2). Therefore, we take inspiration from approaches commonly used for score generative models (Song & Ermon, 2019): We start with a large standard deviation $\sigma_{\max}$ and then reduce it over the optimization process to $\sigma_{\min}$. Since the different EM images can have different levels of magnification, we define the standard deviation depending on the real-world virus size in nm. We choose $\sigma_{\max}$ such that the entire image will be visible if the mask is placed in the center of the image of the smallest magnification level. In practice, exponential decay performed the best when interpolating between $\sigma_{\max}$ and $\sigma_{\min}$. Figure 2 shows an illustration of gradient magnitude and direction at different points in an image for multiple $\sigma_t$.

### 3.3 VIRUS REMOVAL

Then, we iteratively repeat *Initialization* and *Optimization*. However, to prevent the virus detection to converge to the same position, we remove the already detected virus by masking it with a circular shape using the known virus size.

### 3.4 STOPPING CRITERIA

To stop the iterative detection, we consider three criteria: 1) During the *Initialization* step we compute the CAM and stop the virus detection when the computed CAM does not show any focus, meaning the minimum value equals the maximum value of the CAM. 2) After applying the *Virus Removal* step, we forward the image through the classifier. If the output score is smaller than a threshold $t$ we stop searching for viruses in the image, as the classifier predicts no remaining viruses in the image. The value of $t$ is chosen based on the smallest threshold used for computing the Mean Average Precision (mAP) metric. 3) During the *Detection* step, we test if the detected region actually contains a virus. We test this by masking everything in the image but the last virus detected and process this image with the pre-trained classifier.

### 3.5 POSTPROCESSING

Once all particles have been detected, we apply non-maximum-suppression, similar to the Faster-RCNN (Ren et al., 2015), to discard low-scoring virus particles that overlap with higher-scoring ones and exploit the fact that virus particles do not overlap in the image plane. Lastly, a bounding box is created for each virus detected in the image, using the known size of the virus. Moreover,

we compute a score for each bounding box by masking all other detected viruses in the image with circular disks and forwarding it to the pre-trained classifier.

## 4 EXPERIMENTS

In this section, we describe the experimental setup used to validate our method. To analyze our results on a variety of viruses, we have focused our experiments on the following five virus types: Herpes virus, Adeno virus, Noro virus, Papilloma virus, and Rota virus. Below, we will briefly discuss these with respect to imaging-relevant virus properties and data availability, before providing details on the conducted user study and discussing the obtained results.

### 4.1 DATA

**Herpes virus.** The Herpes virus causes lifelong infections in humans. It is composed of an icosahedral capsid with double-stranded DNA, a tegument (middle layer), and an outer lipid bilayer envelope. We use the data from Shaga Devan et al. (2021) which contains 359 EM images with 2860 annotated bounding boxes of the virus particles in total. We use 287 images for training, 36 for validation, and 36 for testing. To approximate the size of the virus, we use values reported by Weil et al. (2020) and Yu et al. (2011) adjusted to account for the different image modality (Read et al., 2019) of room temperature Transmission Electron Microscopy (TEM). This results in a virus size of 165nm, which is also the average size in the data set.

**Adeno virus.** The Adeno virus is a non-enveloped icosahedral capsid with dsDNA. It can infect the lining of the eyes, airways and lungs, intestines, urinary tract, and nervous system leading to cold-like symptoms. We use the data from Matuszewski & Sintorn (2021) containing 67 negative stain TEM images of the Adeno virus with location annotations. We approximate the virus size with 80nm as reported in literature by Goldsmith & Miller (2009).

**Noro virus.** The Noro virus is a small-sized, non-enveloped capsid with icosahedral geometries and single-stranded RNA. It can cause acute gastroenteritis. For this virus, we use 54 negative stain TEM images from Matuszewski & Sintorn (2021) with location annotations. We approximate the virus size with 30nm as reported in literature by Ludwig-Begall et al. (2021).

**Papilloma virus.** The Papilloma virus is a common virus in humans. While it can cause small benign tumors, it can also progress to cervical cancer in high-risk forms. It is non-enveloped with icosahedral DNA. Here, we use the data from Matuszewski & Sintorn (2021) containing 31 negative stain TEM images of the Papilloma virus with location annotations. We approximate the virus size with 50nm as reported in literature by Doorbar et al. (2015).

**Rota virus.** The Rota virus has a distinctive wheel-like shape: round with a double-layered capsid, non-enveloped, with double-stranded RNA (segmented RNA genome). We use the data from Matuszewski & Sintorn (2021) containing 36 negative stain TEM images of the Rota virus with location annotations. We approximate the virus size with 75nm as reported in literature by Yates (2014).

It can be noted that the data sets of the Adeno, Noro, Papilloma and Rota virus (maximum of 67 images) are significantly smaller than the data set of the Herpes virus (359 images). For all viruses, we work on image patches with a resolution of $224 \times 224$ pixels following the standard image input size for state-of-the-art image classifiers. To generate the patches we use a sliding window with no overlap.

### 4.2 USER STUDY

We conducted a user study to compare the cost of obtaining different types of annotations, such that we later can analyze detection accuracy in relation to spent annotation time. The three types of annotation we collect during this study are 1) binary labels indicating virus presence, 2) bounding boxes that precisely describe the virus location and extent, and 3) locations of the virus centers, which

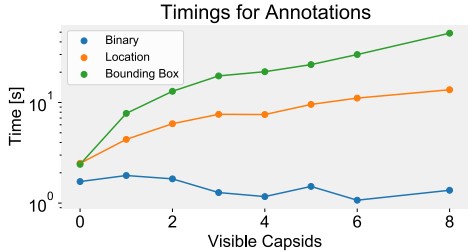
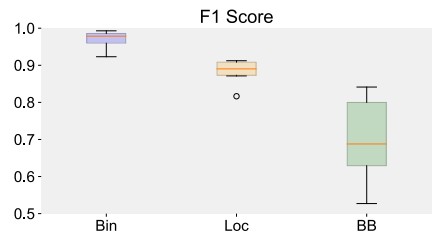

Figure 3: Annotation time vs. the number of particles in the image. Note the log scale.

Figure 4: User Study results for different types of annotations.

is another popular choice for collecting weak detection labels (Li et al., 2019; Matuszewski & Sintorn, 2018; 2019).

**Study design.** During the study, six experts were asked to annotate 85 patches of the TEM images of the Herpes virus with the three types of annotations. We provided an in-house application with a user interface designed to maximize the annotation speed of the three types of labels. We permuted the order of the conditions presented to the experts by a balanced Latin square. We presented the same data to all our participants during all conditions. To counterbalance a learning effect, we randomized the order of the presented patches. The 85 TEM patches show a range of 1-8 visible virus capsids, which is the full range of visible Herpes capsids in the data. For more information and results see the appendix.

**Task performance.** We compare the performance of the participants in all three different tasks. We consider the $F_1$ score as the performance metric. For the evaluation of the location and bounding box task, we use an IoU threshold of $0.5$ to define a True Positive (TP). We found significant differences in the performance of the participants between all tasks (see Figure 4). This reveals the increasing complexity of annotating binary labels, location labels, and bounding box labels: While binary annotations only require the decision of whether a virus is present in an image or not, localization and bounding box annotations require this decision for every visible particle in the image, making the annotations more prone to errors. Additionally, the bounding box annotations require the definition of the size of a virus, thus increasing their complexity even more. These results support the motivation for using binary labels, as the annotations are less prone to errors.

**Annotation times.** Moreover, we investigate the annotation times per visible virus for all tasks. Figure 3 shows the average annotation times per patch of visible virus capsids. The time was measured between the moment showing the image and the user interaction to trigger the visualization of the next image. The average annotation times are slightly decreasing for the binary annotations when the number of visible virus capsids increases. We assume this to be the case based on a simpler detection of virus capsids when their occurrence is higher. However, for both other conditions, the annotation times increase with the number of visible virus capsids. This is accountable to the need for an independent annotation for every single virus.

### 4.3 EXPERIMENTAL SETUP

Our experiments include a wide range of object detection models with different levels of supervision. First of all, we include a fully supervised object detection model (BB). Second, we follow Li et al. (2019) and Matuszewski & Sintorn (2018; 2019) to use minimal labels for training an object detection model (Loc). We derive the bounding boxes for training from location labels and set their sizes by the known virus size. Finally, we compare the bounding boxes resulting from our optimization process (Ours(Opt)) and an object detection model trained with such boxes (Ours(OD)). We use a ResNet-101 (He et al., 2016) as our classification model and a Faster-RCNN with a ResNet-101 backbone as our object detection model. Additionally, we adapt two recent zero-shot segmentation models, SAM and CutLER, to work in a weakly supervised setup: We pick images of the training set that contain a virus and forward these through the pre-trained models to generate bounding boxes. We then find a suitable range of bounding box sizes on the validation set. The resulting range is used in the train set to obtain bounding boxes that are later used to train an object detection model.

Next, we compare against different weakly supervised methods: GradCAM Selvaraju et al. (2017), LayerCAM Jiang et al. (2021), TS-CAM Gao et al. (2021) and Reattention Su et al. (2022). As current state-of-the-art methods found that ViT-based architectures can be beneficial for WSOL, we compare the use of a ViT-B/16 backbone and a ResNet-101 backbone for GradCAM as well as LayerCAM. For methods that rely on saliency maps, we compute multiple bounding boxes by thresholding the maps and deriving bounding boxes from connected regions using Bolelli et al. (2019). We choose the threshold based on the best result on the validation set and apply it to the test set. For a more fair comparison, we also include the knowledge about the virus size in the compared approaches. We report the best results over several runs with different hyperparameters. An extensive evaluation can be found in the appendix A.5.

In all experiments, we obtained three runs with different seeds and reported the mean and standard deviation. For all methods, we perform a parameter search to find the best hyper-parameters. To measure the performance of the object detection models, we use mean average precision with an overlap of $50\,\%$ ($mAP_{50}$).

### 4.4 RESULTS

To compare the three different types of annotations, we fix an annotation time budget and pick random image patches until the time is exhausted. We define the budget as the total time required to annotate the entire data set using binary labels. To compute the time cost of each image, we average the annotation times of the experts participating in the user study. Table 1 presents the results of this experiment. It can be observed that our method is able to outperform location and bounding box labels for all viruses. In particular, $\mathrm{Ours(OD)}$ is outperforming all other approaches. Moreover, we can see that location labels are not able to perform well for some of the virus particles due to the small number of training images.

In our comparison to different zero-shot learning approaches, we found that, in the case of negative stain TEM images where the background is stained making the virus the most prominent structure, both SAM and CutLER performed comparably well. However, when dealing with small viruses such as Noro and Papilloma, CutLER's performance was subpar, while SAM struggled particularly with the smallest virus, Noro. In general, the performance of these methods heavily depends on the sample, noise levels, and preparation method. Our approaches, on the other hand, are more stable over all data sets, leading to the best results on all viruses except for the Adeno.

In conclusion, our investigation revealed that existing weakly supervised methods faced challenges in effectively detecting viruses in EM. Despite incorporating supplementary information about virus size into all comparison approaches, their performance remained suboptimal. This limitation is likely attributed to the fact that contemporary state-of-the-art methods thrive on large data set sizes, which are not available for the detection of virus capsids in EM. Furthermore, these methods are typically crafted to excel in scenarios with more object-centric data sets, whereas EM images present a distinctive challenge by containing numerous object instances within a single frame. Moreover, the conventional methods are not inherently equipped to handle the low signal-to-noise ratio (SNR) characteristic of EM. The prevalence of low SNR introduces inherent ambiguity in object boundaries, a challenge that can be partially mitigated by incorporating the known virus size into the methods, thereby circumventing this ambiguity. Additionally, the presence of noise and low-contrast regions in EM images poses obstacles to extracting discriminative features crucial for precise object localization. This became evident in certain classifiers trained on negative stain TEM data, leading to a bias towards virus borders (see Figure 9). Our introduced optimization, involving a fixed size, contributes to a more robust localization, addressing these challenges.

These findings underscore the pressing need for methodologies purposefully tailored to excel in the intricate task of virus detection in EM. The unique characteristics of EM data necessitate specialized approaches that can navigate the challenges posed by low SNR, small data set sizes and the abundance of object instances within a single image.

**Reduced Annotation Time.** We further investigate the impact of reducing the annotation times. For this experiment, we choose the herpes virus as it has the largest amount of annotated images as well as bounding box annotations. According to our study, this data set requires an annotation time of 19 hours to annotate bounding boxes, 17 hours to annotate location labels and 11 hours to annotate binary labels. We use the time required to annotate all available images using our binary labels as the upper bound of $100\,\%$ and reduce annotation times to $75\,\%$, $50\,\%$, $25\,\%$, $10\,\%$, and $5\,\%$.

Table 1: Comparison of the different methods for the different viruses reporting mAP$_{50}$.

|  | Herpes | Adeno | Noro | Papilloma | Rota |
|---|---|---|---|---|---|
| • BB | 89.18 ±0.95 | - | - | - | - |
| • Loc | 88.13 ±0.38 | 26.24 ±19.93 | 00.82 ±0.34 | 27.20 ±16.44 | 06.51 ±4.66 |
| • Ours(Opt) | 86.98 ±1.92 | 47.85 ±11.82 | 54.65 ±4.94 | 70.02 ±2.85 | 71.73 ±3.51 |
| • Ours(OD) | **91.20** ±0.24 | 58.28 ±5.91 | **74.32** ±1.18 | **78.33** ±2.40 | **78.34** ±2.15 |
| SAM | 41.34 ±4.60 | 44.62 ±3.90 | 08.80 ±3.92 | 73.23 ±7.02 | 66.71 ±4.33 |
| CutLER | 64.95 ±1.98 | **68.49** ±5.44 | 10.72 .±5.63 | 23.02 ±6.73 | 75.5 ±1.44 |
| GradCAM $_{ResNet}$ | 78.79 ±2.04 | 19.17 ±0.78 | 05.54 ±2.99 | 11.57 ±4.17 | 31.78 ±21.58 |
| LayerCAM $_{ResNet}$ | 78.44 ±2.73 | 16.48 ±9.34 | 05.04 ±1.91 | 10.87 ±5.33 | 31.22 ±20.07 |
| GradCAM $_{ViT}$ | 61.87 ±11.87 | 08.00 ±2.12 | 19.31 ±13.64 | 04.03 ±4.52 | 13.12 ±7.37 |
| LayerCAM $_{ViT}$ | 68.33 ±6.59 | 09.18 ±5.64 | 10.82 ±11.78 | 17.41 ±11.33 | 09.74 ±2.42 |
| TS − CAM | 32.06 ±1.02 | 39.25 ±4.13 | 14.64 ±4.66 | 07.11 ±3.85 | 43.53 ±3.93 |
| Reattention | 68.85 ±0.62 | 58.49 ±2.22 | 55.09 ±8.92 | 35.60 ±13.01 | 59.05 ±11.40 |

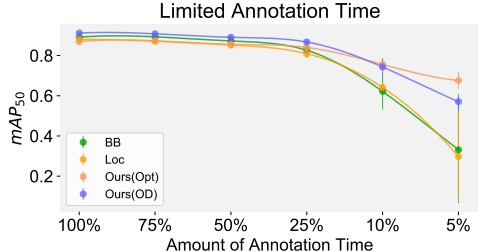

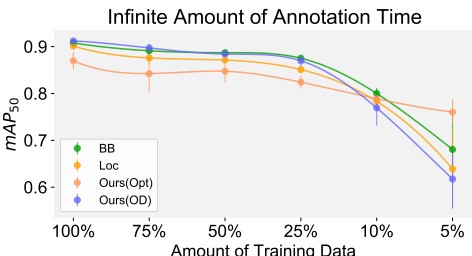

Figure 5: Comparison of a detector model using different annotation times.

Figure 6: Comparison of a detector model using different data set sizes.

Figure 5 presents the results of this experiment. We found that 1) Ours(OD) is able to outperform other types of labels for all the time budgets. 2) Ours(Opt) can provide better performance than all other methods, including Ours(OD), when the data set is small (annotation time less than 25 %).

**Infinite Annotation Time.**    Moreover, we evaluated the performance of all four methods when an infinite time for annotation is possible, but the amount of data is limited. The results are presented in Figure 6. We here elaborate again on the results of the Herpes virus, as it has the largest amount of annotated images as well as bounding box annotations. However, we also include results on the additional virus data sets in the appendix Table 6. It can be observed that Ours(OD) obtained similar or slightly better performance than Loc and BB when the data set size is large. Moreover, we can see that Ours(Opt), although not able to reach the performance of the other methods, is able to achieve competitive performance. However, for small data set sizes, we see that the supervised approaches start to outperform the weakly supervised approach. We believe that this has two reasons: First, the smaller data set sizes do not allow to train a classifier, with good localization abilities. Additionally, training the Faster-RCNN on a data set that is small and noisy leads to worse performance. However, please note that the benefit of annotating image-level labels vanishes as the absolute time for annotation is already small.

## 5 CONCLUSION

In this paper, we proposed a novel approach for virus particle detection in EM data based on weak supervision. Our approach optimizes bounding box positions of virus particles by leveraging a pre-trained classifier, Gaussian masking and domain-specific knowledge. Furthermore, to improve the optimization, we initialize the Gaussian masks based on GradCAM hotspots. We compared the results obtained with our method to other weakly supervised approaches, as well as fully supervised ones, where we show that our method is able to outperform those for the same amount of annotation time. Moreover, we conducted a user study that shows that binary labels are easier to obtain and more robust against errors than other annotation methods. Thus, our approach shows promise for efficient and accurate particle detection in EM images, opening new avenues for practical applications in this field. In the future, we would like to analyze the applicability of our method to the localization of objects that vary in size.

ACKNOWLEDGMENTS

We would like to thank *Jens von Einem* (Institute of Virology, Ulm University Medical Center) for providing herpesvirus infected cells.
This work was financed by the Baden-Württemberg Stiftung (BWS) for the ABEM project under grant METID12–ABEM.

REPRODUCIBILITY STATEMENT

The source code associated with the experiments conducted in this paper is publicly available on GitHub at the following link: https://github.com/HannahKniesel/WSCD. Instructions for replicating experiments presented in the paper will be provided in the repository. This includes information on command-line arguments, hyperparameters, and any additional configurations necessary to reproduce the reported results.

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

# A   APPENDIX

## A.1   ADDITIONAL EXPERIMENTS

In this section we present additional ablation studies. All our experiments were done using the herpes virus and results are reported on the validation set for $\mathrm{Ours(Opt)}$.

### A.1.1   OPTIMIZATION

While we investigate the impact of the initialization for the optimization in our main paper (Section 4.5), Table 2 shows results obtained when using a different number of iteration steps for the optimization. As can be seen, our initialization not only reduces the number of needed forward passes, but also outperforms all other initialization schemes for all numbers of iterations. Based on these findings, we have chosen a maximum number of iterations during optimization.

| Method | Iterations | mAP$_{50}$ | Forward Passes |
|---|---|---|---|
| Random | 0 | $2.93 \pm 0.73$ | $5 \pm 0$ |
| | 25 | $6.15 \pm 0.92$ | $117 \pm 6$ |
| | 50 | $11.63 \pm 1.36$ | $179 \pm 7$ |
| | 100 | $36.9 \pm 1.33$ | $289 \pm 19$ |
| | 200 | $60.24 \pm 2.29$ | $518 \pm 14$ |
| Selective Search | 0 | $57.86 \pm 4.94$ | $145 \pm 68$ |
| | 25 | $63.39 \pm 5.75$ | $169 \pm 76$ |
| | 50 | $64.76 \pm 6.68$ | $186 \pm 75$ |
| | 100 | $65.44 \pm 7.49$ | $232 \pm 88$ |
| | 200 | $66.77 \pm 7.28$ | $344 \pm 142$ |
| GradCAM | 0 | $\mathbf{68.66} \pm 3.14$ | $1 \pm 0$ |
| | 25 | $\mathbf{77.58} \pm 1.23$ | $32 \pm 1$ |
| | 50 | $\mathbf{80.35} \pm 0.8$ | $64 \pm 2$ |
| | 100 | $\mathbf{81.4} \pm 0.91$ | $128 \pm 5$ |
| | 200 | $\mathbf{82.4} \pm 0.25$ | $255 \pm 9$ |

Table 2: Comparison of different position initialization methods. Iterations denotes the maximum number of steps used for optimizing a single position. As expected, we found, that a random initialization will require more optimization steps to converge to a good position. An initialization with selective search already reduces the amount of optimization steps needed, while our proposed initialization, based on GradCAM, reduces the needed optimization steps the most, while at the same time requires fewest forward passes.

To conclude, our introduced optimization can converge to the correct position of the virus, even though the initialization of the position is not close to a virus particle. However, combining the optimization with a good initialization scheme, like the proposed GradCAM initialization, the number of iterations can be reduced, saving computation time.

### A.1.2   INITIALIZATION

We evaluate the impact of GradCAM initialization compared to selective search (Uijlings et al., 2013), random initialization, and a greedy version of our optimization approach: the sliding window. For the sliding window, we use all possible positions in the image separated by a distance of $0.125 \times r$ of each other as bounding box candidates. For the selective search, we pick the highest scoring region proposal that has a size bigger than $0.8\times$ virus size and smaller than $1.2\times$ virus size. Lastly, for GradCAM we use the initialization schema introduced in  section 3. Additionally, to compare the computational cost of the approaches, we report the average number of forward passes through the pre-trained network when a virus has been detected. Results can be found in Table 3.

We found that random initialization and selective search are not performing well when compared to GradCAM. The sliding window underperforms when compared to our method since its performance

Table 3: Comparison of different approaches to initialize the bounding box position to optimize.

|  | mAP$_{50}$ | Forward Passes |
|---|---|---|
| Sliding Window | 78.91 $_{\pm 4.73}$ | 3834 $_{\pm 70}$ |
| Random | 60.24 $_{\pm 2.29}$ | 518 $_{\pm 14}$ |
| Selective Search | 66.77 $_{\pm 7.28}$ | 344 $_{\pm 142}$ |
| GradCAM | **82.40** $_{\pm 0.25}$ | 255 $_{\pm 9}$ |

Table 4: Evaluation of the standard deviation of the Gaussian mask.

|  | mAP$_{50}$ |
|---|---|
| $2 \times r$ | 68.68 $_{\pm 2.02}$ |
| $1 \times r$ | 75.66 $_{\pm 3.24}$ |
| $0.5 \times r$ | **75.75** $_{\pm 2.60}$ |
| $0.25 \times r$ | 75.36 $_{\pm 2.83}$ |

Table 5: Comparison of different backbones for our approach reporting mAP$_{50}$.

|  | Herpes | Adeno | Noro | Papilloma | Rota |
|---|---|---|---|---|---|
| Ours(Opt) ResNet | **86.98** $_{\pm 1.92}$ | **47.85** $_{\pm 11.82}$ | **54.65** $_{\pm 4.94}$ | **70.02** $_{\pm 2.85}$ | **71.73** $_{\pm 3.51}$ |
| Ours(Opt) ViT | 48.66 $_{\pm 07.44}$ | 07.46 $_{\pm 05.21}$ | 10.44 $_{\pm 09.06}$ | 05.67 $_{\pm 08.05}$ | 04.50 $_{\pm 3.51}$ |

highly depends on the distance between consecutive positions. When comparing the number of forward passes, we can see that the sliding window approach requires the most forward passes of the classifier. Moreover, we can see that selective search requires fewer forward passes than random initialization, indicating that it converges faster. GradCAM requires the least forward passes while obtaining the best performance.

### A.1.3 GAUSSIAN STANDARD DEVIATION

We investigate the influence of $\sigma_{\min}$ on the Gaussian mask. Given the virus radius $r$, we found that $\sigma_{\min} = 0.5 \times r$ gives the best results (see Table 4).

### A.2 BACKBONE ARCHITECTURES

We found that most state of the art approaches use ViT (Dosovitskiy et al., 2020) backbones to achieve superior performance. We hence compare the use of a ViT-B/16 backbone compared to a ResNet-101 backbone. We found, that given the small virus data sets ViT backbones were not able to perform well (see Table 5).

### A.3 INFINITE ANNOTATION TIME

In line with the paragraph *Infinite Annotation Time* (Section 4.4) in the main paper, we evaluate our approach when all available data is needed and hence the annotation times are not balanced. For results see Table 6.

In line with our findings in the main paper in Section 4.4, fully supervision is outperforming our weakly supervised approach when the data set sizes become small, as it is the case for the Adeno virus, the Noro virus, the Papilloma virus and the Rota virus. For large data set sizes (the Herpes virus) the annotation of more complex labels like location annotations or bounding boxes do not improve the result.

**Discussion.** Considering limited amount of available data, the experiment with the herpes data set suggests, that the annotation of image level labels is beneficial in two cases: First, when the data set is too small to sufficiently train a detector model, our pseudo labels, Ours(Opt), that do not rely on the training of state of the art detector models, but only require a well performing classifier, outperform object level annotations. Second, when the data set is large enough our pseudo labels are performing well enough to train a state of the art object detection model, Ours(OD), that is en par with a similar model that has been trained on object level annotations, suggesting that annotation time can be reduced when only annotating image level labels instead of object level labels.

However, for the 10% and 25% split of the Herpes virus, as well as the Adeno virus, Noro virus, Papilloma virus and Rota virus, bounding boxes or location labels seem to be beneficial. We believe this to be the case, since the size of the data set is sufficient to train a detector model. However, small

Table 6: Comparison of the different methods for the different viruses reporting mAP$_{50}$. We here evaluate the approaches for limited data, instead of limited annotation time.

|  | Herpes | Adeno | Noro | Papilloma | Rota |
|---|---|---|---|---|---|
| • BB | 90.77 $\pm$0.99 | - | - | - | - |
| • Loc | 90.11 $\pm$0.48 | **80.25** $\pm$3.54 | **89.43** $\pm$1.12 | **91.74** $\pm$0.47 | **89.51** $\pm$1.76 |
| • Ours(Opt) | 86.98 $\pm$1.92 | 47.85 $\pm$11.82 | 54.65 $\pm$4.94 | 70.02 $\pm$2.85 | 71.73 $\pm$3.51 |
| • Ours(OD) | **91.20** $\pm$0.24 | 58.28 $\pm$5.91 | 74.32 $\pm$1.18 | 78.33 $\pm$2.40 | 78.34 $\pm$2.15 |

errors in the pseudo labels can cause the detector to perform worse, based on the rather small training data sets. This means, that the performance of our method heavily depends on the performance of the classifier. Qualitative results (Figure 17) on the virus data sets suggest that our pseudo labels (Ours(Opts)) are prone to detect False Positive (FP). While their score is low, the Faster-RCNN (Ours(OD)) is able to differentiate between TP and FP. However, when the data set size is sufficiently big to train a Faster-RCNN, bounding box labels do not share this disadvantage and result in a better performing Faster-RCNN. With increased data set sizes, the detector will perform better and better. Additionally, as mentioned above, the error of the pseudo labels gets smaller, since the location ability of the classifier increases, and the large amount of data is more forgiving for small errors. This finally results in Ours(OD), performing en par with methods that rely on object level annotations.

However, we argue that the most limiting factor is not the availability of data but the annotation time required to generate labels, which is why we emphasize on the results of the *limited annotation times*. Raw EM data is available in large amounts, for example in data bases such as EMPIAR (Iudin et al., 2016). This data base has grown immensely (Iudin et al., 2016) based on the current research topics in bio-medicine. Despite this, the data does not come with annotation labels. Results in our main paper suggest, that when the limiting factor is the annotation time, one should annotate image level labels and use our suggested approach to optimize for the position of the virus particles. Additionally, we would like to point out, that when the data set sizes become small, the benefit of annotating image level labels vanishes, as the absolute time for annotation is already quite small.

## A.4 INFLUENCE OF APPROXIMATED OBJECT SIZE

As our method additionally requires the knowledge of the objects size, we conduct an experiment on the influence of the error in the objects size. We therefore corrupt the known object size by 10%, 20% and 30%. As expected, we found a decrease in performance with increased error of the objects size (see Table 7). However, we would like to highlight, that in the case of virus detection, the sizes can most often be retrieved from literature. Additionally, an accurate approximation of the virus size can be retrieved from measuring only a few instances as the standard deviation of the virus size can be neglected.

Table 7: Influence of error in size approximation of the object reporting mAP$_{50}$.

| Error | 0% | 10% | 20% | 30% |
|---|---|---|---|---|
| Ours(Opt) | **86.98** $\pm$1.92 | 82.38 $\pm$02.34 | 79.14 $\pm$03.10 | 67.31 $\pm$02.03 |

## A.5 WEAKLY SUPERVISED COMPETITORS USING VIRUS SIZE

To be able to make a more fair comparison to other state of the art approaches, we include the known virus size into existing methods. We test two different ways of doing this and also check their combination. First, we try to remove noise by filtering detected boxes based on the known virus size. Therefore, we define a size range $r$ based on the virus size $v$, such that the size of the detected boxes falls in the range of $\in [(1 - r) \times v, (1 + r) \times v]$. Next, we additionally replace the detected boxes with boxes of the known virus size. Results can be seen in Table 8.

Table 8: Comparison of different ways to incorporate the known virus size into existing weakly supervised approaches reporting mAP$_{50}$. We incorporate filtering the detected bounding boxes by the virus size (F) and replacing the detected boxes with boxes of the actual virus size (R).

| | F | R | Herpes | Adeno | Noro | Papilloma | Rota |
|---|---|---|---|---|---|---|---|
| GradCAM ResNet | ✗ | ✗ | 57.28 ±3.15 | 12.21 ±8.05 | 01.96 ±1.02 | 04.70 ±2.35 | 19.20 ±19.64 |
| | ✗ | ✓ | **78.79** ±2.04 | 15.18 ±8.51 | **05.54** ±2.99 | **11.57** ±4.17 | 31.28 ±21.53 |
| | ✓ | ✗ | 61.34 ±2.85 | 14.39 ±1.09 | 02.67 ±2.11 | 06.90 ±3.37 | 28.56 ±18.82 |
| | ✓ | ✓ | 76.60 ±2.57 | **19.17** ±0.78 | 03.99 ±2.58 | 09.20 ±3.27 | **31.78** ±21.58 |
| LayerCAM ResNet | ✗ | ✗ | 56.38 ±3.82 | 09.69 ±6.58 | 01.88 ±0.94 | 05.10 ±2.26 | 26.11 ±19.62 |
| | ✗ | ✓ | **78.44** ±2.73 | **16.48** ±9.34 | **05.04** ±1.91 | **10.87** ±5.33 | **31.22** ±20.07 |
| | ✓ | ✗ | 61.74 ±1.33 | 12.92 ±9.18 | 02.34 ±2.15 | 05.47 ±2.57 | 29.20 ±19.98 |
| | ✓ | ✓ | 76.46 ±4.69 | 16.26 ±11.47 | 03.74 ±2.45 | 08.47 ±4.05 | 29.27 ±20.48 |
| GradCAM ViT | ✗ | ✗ | 44.00 ±15.31 | 00.66 ±0.53 | 08.98 ±9.38 | 01.24 ±1.4 | 03.52 ±3.98 |
| | ✗ | ✓ | 58.52 ±10.39 | **08.00** ±2.12 | **19.31** ±13.64 | **04.03** ±4.52 | **13.12** ±7.37 |
| | ✓ | ✗ | 57.02 ±13.09 | 04.91 ±3.67 | 13.45 ±10.05 | 03.14 ±1.89 | 09.02 ±7.07 |
| | ✓ | ✓ | **61.87** ±11.87 | 06.4 ±4.58 | 16.24 ±13.38 | 02.96 ±2.09 | 09.02 ±7.07 |
| LayerCAM ViT | ✗ | ✗ | 49.22 ±7.96 | 00.50 ±0.27 | 03.51 ±3.99 | 04.65 ±3.46 | 01.02 ±0.17 |
| | ✗ | ✓ | 65.47 ±7.43 | **09.18** ±5.64 | **10.82** ±11.78 | **17.41** ±11.33 | 06.43 ±1.76 |
| | ✓ | ✗ | 61.27 ±7.32 | 02.88 ±3.67 | 06.33 ±6.65 | 06.33 ±6.65 | **09.74** ±2.42 |
| | ✓ | ✓ | **68.33** ±6.59 | 03.67 ±4.78 | 08.62 ±11.26 | 08.62 ±11.26 | **09.74** ±2.42 |
| TS − CAM | ✗ | ✗ | 07.57 ±0.69 | 05.34 ±1.89 | 04.20 ±2.31 | 01.07 ±0.47 | 05.50 ±2.12 |
| | ✗ | ✓ | 18.40 ±1.99 | 11.54 ±2.92 | 11.10 ±3.18 | **07.11** ±3.85 | 16.67 ±6.38 |
| | ✓ | ✗ | 24.69 ±1.21 | 25.03 ±9.56 | 10.94 ±4.74 | 04.03 ±2.01 | 32.48 ±3.29 |
| | ✓ | ✓ | **32.06** ±1.02 | **39.25** ±4.13 | **14.64** ±4.66 | 05.77 ±3.84 | **43.53** ±3.93 |
| Reattention | ✗ | ✗ | 37.93 ±4.3 | 30.81 ±6.12 | 25.41 ±1.99 | 14.61 ±9.82 | 24.52 ±12.64 |
| | ✗ | ✓ | 42.72 ±4.3 | **58.49** ±2.22 | 48.87 ±5.77 | 31.39 ±17.1 | 41.05 ±16.5 |
| | ✓ | ✗ | 65.05 ±1.59 | 42.37 ±10.9 | 44.29 ±2.92 | 28.13 ±12.9 | 45.96 ±5.85 |
| | ✓ | ✓ | **68.85** ±0.62 | 57.58 ±1.32 | **55.09** ±8.92 | **35.6** ±13.01 | **59.05** ±11.4 |
| Ours(Opt) | | | 86.98 ±1.92 | 47.85 ±11.82 | 54.65 ±4.94 | 70.02 ±2.85 | 71.73 ±3.51 |
| Ours(OD) | | | **91.20** ±0.24 | **58.28** ±5.91 | **74.32** ±1.18 | **78.33** ±2.40 | **78.34** ±2.15 |

For an easier comparison, we also include the results of our method in the table. Here, the virus size is used like explained in the main paper.

We found that incorporating the size into other weakly supervised methods leads to a strong performance increase. In general, Reattention was performing the best out of all other weakly supervised approaches. Still, our methods were mostly able to outperfom other approaches by a large margin. For a more in the depth discussion we refer to the main paper subsection 4.4.

|              | Herpes | Adeno | Noro  | Papilloma | Rota  |
| ------------ | ------ | ----- | ----- | --------- | ----- |
| DoG + response | 18.16 | 15.09 | 15.53 | 21.78 | 21.33 |
| DoG          | 14.27  | 09.54 | 08.44 | 18.62     | 02.89 |

Table 9: Standard computer vision does not require the annotation of large data sets making it especially feasible for EM. Additionally, we can inform the process with the known virus size, further pushing its performance. However, we found that due to the high levels of noise DoG features fail to reliably detect keypoints.

Table 10: Evaluation of the loss computation.

| Loss   | $mAP_{50}$ |
| ------ | ---------- |
| Score  | 75.75 $\pm2.60$ |
| Logits | **76.72** $\pm1.62$ |

Table 11: Evaluation of the masking strategy.

| Masking    | $mAP_{50}$ |
| ---------- | ---------- |
| Zeros      | 72.01 $\pm2.32$ |
| Inpainting | 75.70 $\pm1.70$ |
| Mean       | **76.72** $\pm1.62$ |

Table 12: Evaluation of the score computation.

| Masking    | $mAP_{50}$ |
| ---------- | ---------- |
| Mask Bkg.  | 76.72 $\pm1.62$ |
| Mask Other | **80.35** $\pm0.80$ |

Table 13: Evaluation of Gaussian masking.

| Loss   | $mAP_{50}$ |
| ------ | ---------- |
| PDF    | **76.22** $\pm 3.23$ |
| no PDF | 75.87 $\pm 2.29$ |

## A.6 VIRUS DETECTION USING DoG

As annotated EM data is limited, another way for virus detection is the use of standard computer vision like Differences of Gaussians (DoG) for object detection. This method only requires a small validation set to find suitable parameters. However, in our experiments we found that DoG is not able to reliably detect virus particles in noisy EM data. For keypoint detection we follow Lowe (2004). Additionally, we use the virus size to define a range of valid keypoint sizes to filter keypoints. We tune the contrast threshold as well as the size range on the validation data set and report the numbers on the test data set. We tried two variants. One uses the maximum score for all detected keypoints and the second uses the keypoint response as the score of the bounding box. We got the best results by normalizing the keypoint response with the highest response in the data set. Results can be found in Table 9. We report $mAP_{50}$ similar to the main paper.

We found that DoG did not perform well to detect the virus particles. We believe that this has multiple reasons: Depending on the state of the virus, the contrast of the capsid varies. Finding a suitable contrast threshold is therefore not trivial. Additionally, there are virus capsids that are hardly visible (based on lower contrast and noise). Lastly, we believe that DoG features by themselves are not well suited to work on EM images, as these images are usually quite noisy. Using machine learning to solve the problem is a more robust choice, first, to also detect virus particles that are not dominant, and second, to transfer the method between different EM-modalities (which can come with different levels of noise), as the features are learned, rather than hand-crafted.

## A.7 ABLATIONS

In this section, we present additional ablation studies.

**Gaussian Masking Ablation**    In our main paper, we conduct all experiments by using the Gaussian PDF, since we found this to work better in practice. Table 13 shows the obtained numbers for a direct comparison of PDF and no PDF, based on which we made this decision.

**Loss Computation**    We compare the use of the classifier's logits to the classifier's score as input to the loss function for the bounding box optimization. We found that when using the logits, gradients can be propagated more favorably (see Table 10).

**Masking Strategy**    We evaluate different masking strategies: masking with zeros, masking with the mean of the data set, and an inpainting technique that pastes the content of a background patch (Dvornik et al., 2018; Dwibedi et al., 2017; Yun et al., 2019). We found that masking with the mean provides the best results (see Table 11).

**Score Computation**    We evaluate different ways to compute the score of the resulting bounding box. For one, we mask everything but the detected virus and forward this through the pre-trained classifier to retrieve the score. Second, we mask all other detected virus particles and forward the masked image through the classifier. We found that the second approach works better (see Table 12).

# B    ADDITIONAL INFORMATION ON DATASET DISTRIBUTION

We provide an in depth overview of the data set distribution that have been used in the main paper.

| Annotation time | 100% | | | 75% | | | 50% | | |
| Amount | Bin | Loc | BB | Bin | Loc | BB | Bin | Loc | BB |
| --- | --- | --- | --- | --- | --- | --- | --- | --- | --- |
| Images | 22987 | 14306 | 12936 | 17240 | 10725 | 9656 | 11497 | 7172 | 6473 |
| Virus | 2235 | 1391 | 1248 | 1693 | 1050 | 957 | 1122 | 669 | 621 |

| Annotation time | 25% | | | 10% | | | 5% | | |
| Amount | Bin | Loc | BB | Bin | Loc | BB | Bin | Loc | BB |
| --- | --- | --- | --- | --- | --- | --- | --- | --- | --- |
| Images | 5748 | 3607 | 3294 | 2303 | 1444 | 1323 | 1149 | 718 | 649 |
| Virus | 571 | 309 | 287 | 202 | 121 | 111 | 102 | 66 | 61 |

Table 14: Absolute numbers of the herpes data set as they have been used in Table 1 and  Figure 5 in the main paper.

# C    USER STUDY

## C.1    MENTAL EFFORT

We additionally investigate a subjective measure of cognitive load by asking for the difficulty of the task (Ayres, 2006) as well as the mental load (Paas et al., 2003) the participant experienced during the task. Right after the completion of each task (binary, location, and bounding box annotation), the participants were asked to rate their mental load as well as the difficulty of the task based on a nine-point Likert scale. Since the participants of our study were well-trained biologists, they found the recognition of virus particles not challenging. Therefore, the differences in difficulty between the tasks were small, being only significantly different between the binary and bounding box labels. However, even if the tasks were not difficult for the participants, their mental load after the annotation process was different for different types of annotation, since their complexities are different.  It requires more effort to locate all viruses on the image than simply identify if there is a virus, and even more effort to draw the enclosing bounding box of all viruses. For mental effort, we were able to see a significant difference in mental load between all types of annotations, binary, location, and bounding boxes.

## C.2    STUDY DESIGN

Here we provide additional details regarding our user study setup. During our user study, six experts were asked to annotate 85 patches of TEM images of the Herpes virus with the three types of annotations: 1) Binary annotations (Bin), indicating the presence of virus particles. 2) Location

annotations (Loc), indicating the center of every virus particle. 3) Bounding boxes (BB), showing the size and position of every virus particle. The participants got paid 10€ per hour.

Every condition was structured into a training phase and a main phase: During the training phase, the participants were asked to read short instructions on how to use the tool.

For the Binary condition the following instruction was presented:

> When you see a virus capsid, use the up key to mark that there is a virus in the image. When you don't see a virus, use the down key to mark that there is no virus. Once you press Up or Down, the next image will be shown.

For the Location condition the following instruction was presented:

> When you see a virus capsid, use the mouse to mark the center of the capsid. Note: not all images will contain a capsid. If there are multiple capsids visible, make sure you mark the location of all capsids. If you click right on a location, the location will be removed. If you click, hold and move with the left mouse, the location will be moved. To get to the next image, use the right arrow key. If there is no virus visible in the patch, you can skip to the next image.

For the Bounding Box condition the following instruction was presented:

> When you see a virus capsid, use the mouse to draw a bounding box around the virus capsid by pressing the left mouse and holding. Note: not all images will contain a virus. If there are multiple virus capsids visible, make sure you draw a bounding box for each of the visible virus capsids. To move an existing box left click (inside the bounding box that is supposed to be moved), hold and move with the mouse. To remove a bounding box right click inside the according bounding box. To get to the next image, use the right arrow key. If there is no virus visible in the patch, you can skip to the next image.

Based on these instructions, the participants were asked to test every functionality of the tool by annotating schematic images (see Figure 7). The participants then labelled 25 TEM images, again extracted from the Herpes virus data, for training purposes. We made sure, that none of the training images were used during the actual annotation task. As a last step of the training phase the participants were asked if they felt comfortable using the tool. All answered with "yes". During the main phase, the participants annotated 85 TEM images. Finally, they were asked to report their mental load as well as the task difficulty using a nine point Likert scale.

As stated in the main paper (Section 4.2), we permute the order of the three conditions by a balanced Latin square. To minimize a bias due to the presented data, we reuse the TEM images for all three conditions. To then counterbalance a learning effect on the presented data, we randomize the order of the TEM images during every condition.

## C.3 ANNOTATION TOOL

The annotation tool was designed in a similar fashion for labeling the bounding boxes, location labels or binary labels (see Figure 8):

Once the participants were ready, they could start the labeling with a click on a button. Then, the first image to annotate was shown. For the *bounding box* annotations, the participants were able to draw bounding boxes using the mouse. These bounding boxes could be edited by changing the size and the position. Once the participants were finished, the right key could be used to get to the next image to label.

For the *location* condition, the participants were again asked to use the mouse to mark a location by a single click. Using drag and drop, this location could be moved. Again, when the participants were finished with one image to label, they could get to the next image using the right key.

For the *binary* case it was possible to implement the interaction with the labeling tool with the keyboard only. Pressing the up-key was labeling the presented image patch as "presence of virus",

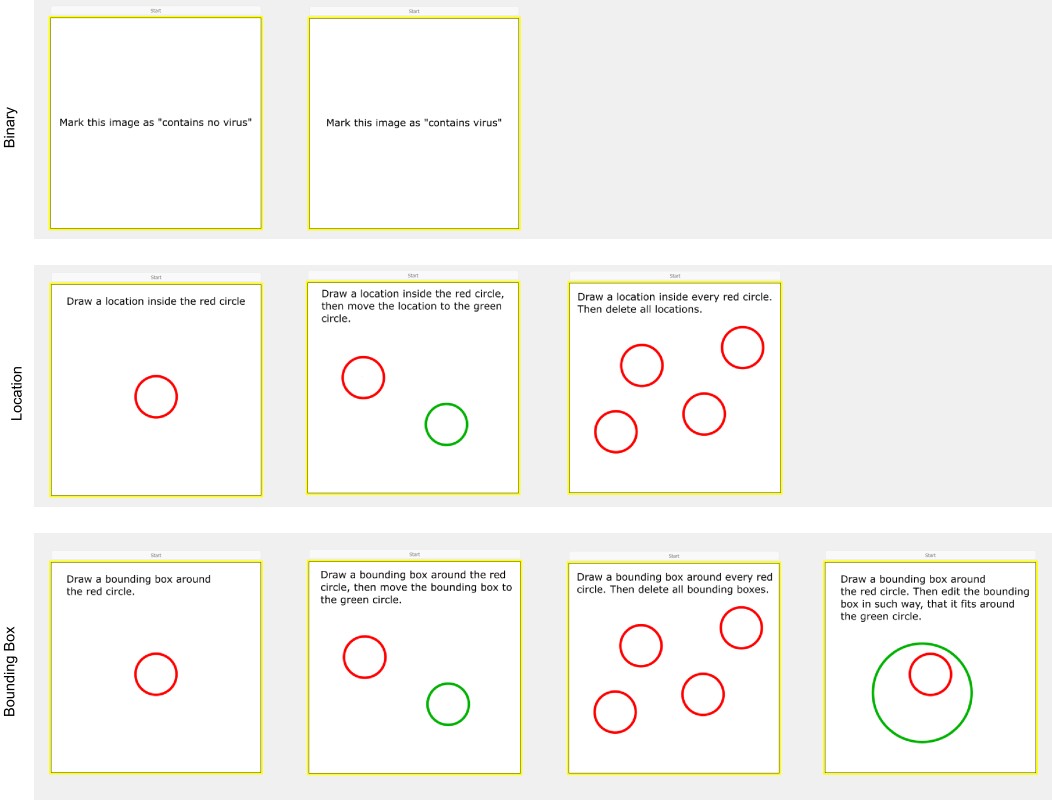

Figure 7: We prepare schematic images to annotate during a small scale training phase of the annotation tool. For each condition, we made sure that the participant tested all functionalities by following the given instructions. Additionally, 25 TEM images are annotated to get familiar with the tool.

while pressing the down-key marked it as "absence of virus". Since this interaction did not require a confirmation of the labelling, the next image to label was presented as soon as either the up or down key was pressed.

For simplicity, the user was not able to go back to the previous image. To make sure that the participants were not able to just skip through the images, we only allowed a "next" click, 0.2 seconds after the image was shown. With 0.2 seconds being the threshold to perceive a single image by a human (similar to frame rates in videos).

## C.4 EVALUATION

As stated in the main paper, for finding significant differences we first conduct a Shapiro Wilk test for normality and use the Levene's test for equal variances. Depending on the outcome, we make use of a paired t-test or the Wilcoxon signed-rank test. We report significant results, when the p-value is smaller than 0.05. For completeness we list our statistical findings here and include further evaluations.

### C.4.1 TASK PERFORMANCE

For the evaluation of the task performance, we consider precision, recall and F1 score. Mean and standard deviation are computed over all participants over all images to annotate. We consider bounding boxes with IoU > 0.5 as TP. For location annotations, we use the known virus radius of 165nm, as described in the main paper in Section 4.1, to derive bounding boxes.

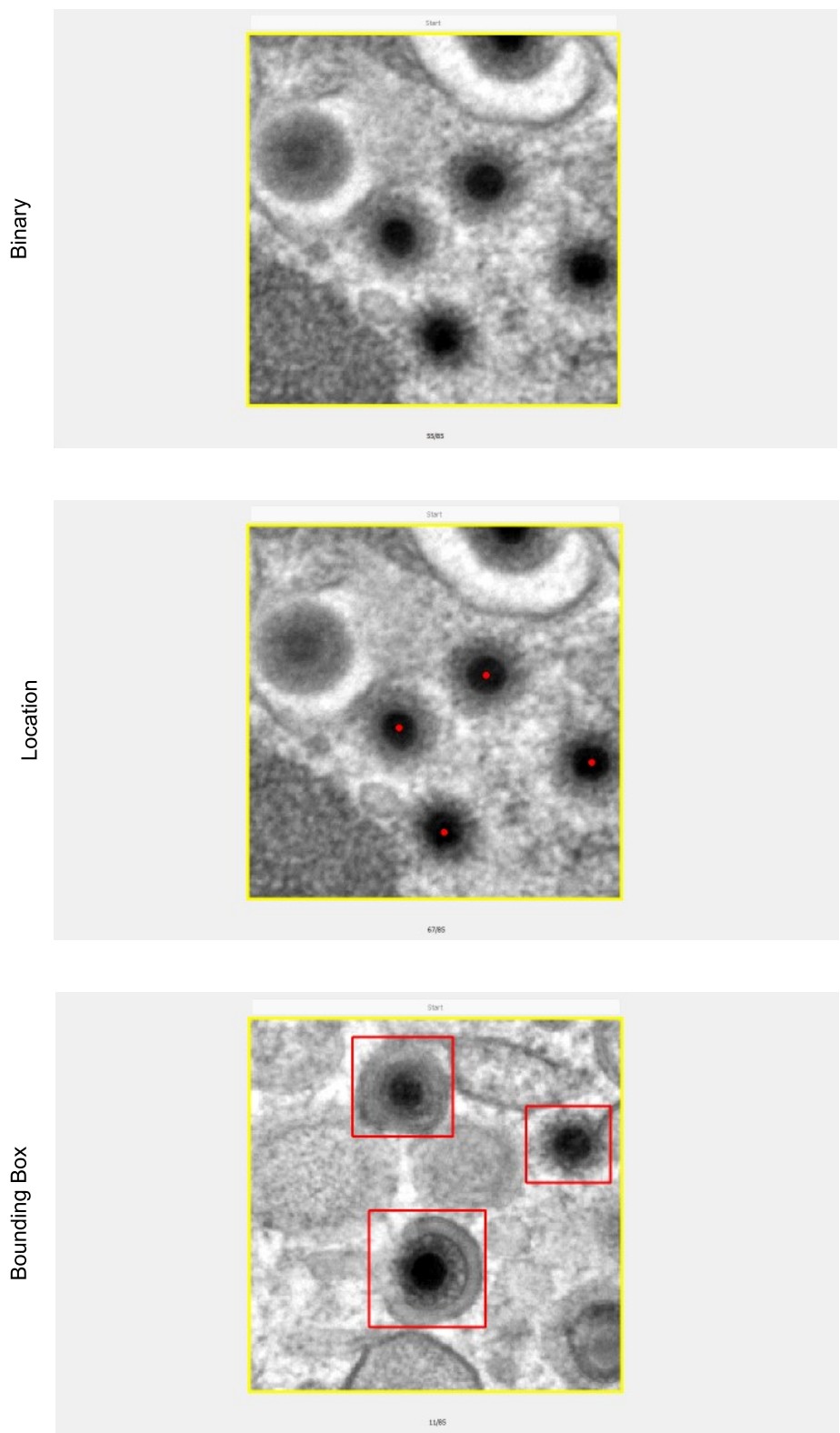

Figure 8: Annotation tool for the three different tasks of annotating binary labels, location labels and bounding boxes.

**Classification**   In a first evaluation, we converted all annotation labels to classification labels. We do so, by classifying all annotations, that contain at least one location or bounding box as "presence of virus", while all other annotations are converted to "absence of virus". We did not find a significant difference between the tasks, when considering this evaluation on classification labels. This allows for the conclusion, that all three tools are equally well designed to classify the images.

Results of the Levene's test can be found in Table 15a. We report the metrics and the outcome of the Shapiro Wilk test for normality in Table 15b. The final statistical results can be seen in Table 15c.

Table 15: We converted all annotations to classification labels and compared the results. We report the p-value of the conducted statistical tests for all metrics. The p-value is the result of a paired t-test or the Wilcoxon signed-rank test, depending on the outcome of the Shapiro Wilk test for normality and the Levene's test for equal variances.

(a) P-value of the Levene's test for equal variances.

|  | Bin - Loc - BB |
|---|---|
| Precision | 0.58 |
| Recall | 0.91 |
| F1 | 0.84 |

(b) We report different performance metrics and run a Shapiro Wilk test for normality for every metric.

|  | Bin | Loc | BB |
|---|---|---|---|
| Precision | $98.83 \pm 1.26$ | $95.72 \pm 5.39$ | $96.45 \pm 6.13$ |
| Shapiro Wilk | 0.0411 | 0.0412 | 0.0009 |
| Recall | $95.24 \pm 5.13$ | $95.18 \pm 3.53$ | $94.75 \pm 4.96$ |
| Shapiro Wilk | 0.1613 | 0.0661 | 0.0056 |
| F1 | $96.9 \pm 2.39$ | $95.3 \pm 2.75$ | $95.37 \pm 3.59$ |
| Shapiro Wilk | 0.2381 | 0.4402 | 0.0380 |

(c) We did not find significant differences, indicating that there is no expected bias due to the annotation tools.

|  | Bin - Loc | Bin - BB | Loc - BB |
|---|---|---|---|
| Precision | 0.2733 | 0.4652 | 0.4652 |
| Recall | 0.9859 | 0.5625 | 0.8927 |
| F1 | 0.2487 | 0.5001 | 1.0 |

**Location**   Second, we convert the bounding box annotations to location annotations and test for significant difference in the performance of the participants. To do so, we compute the center of the bounding box as location and place a bounding box with the known virus size of 165nm. We include mean and standard deviation for precision, recall and F1 score (see Table 16b). Additionally, results of the Levene's test for equal variances can be found in Table 16a. We did not find a significant difference in the performance of the participants (see Table 16c). This allows for the conclusion, that the annotation tools for the location and bounding boxes are equally well designed.

**Comparison**   While the first two evaluations indicate that there is no expected bias due to the annotation tools, this evaluation considers an actual comparison between all three tasks.

We test for equal variances using the Levene's test (see Table 17a). We report precision, recall and F1 score for the performance of the participants annotating binary labels, location labels and bounding boxes, as well as the statistics of the Shapiro Wilk test for normality (see Table 17b). As stated in Section 4.2 in the main paper, we found a significant difference between all three tasks (see Table 17c). For completeness we list our statistical findings here.

**Mental Load**   As described in the appendix, we were also asking the participants to rate their subjective measure of mental load and the difficulty of the task. For completeness, we here show a detailed statistical evaluation, which follows the same scheme as the previous one: First, we test for equal variances with the help of the Levene's test. Next, we test for normality by using the Shapiro Wilk test and report the median of the Likert scales. Finally, we check for significant differences using a paired t-test or the Wilcoxon signed-rank test, depending on the outcomes of the previous tests.

Table 16: We converted all bounding box annotations to localization annotations and compared the results. We report the p-value of the conducted statistical tests for all metrics. The p-value is the result of a paired t-test or the Wilcoxon signed-rank test, depending on the outcome of the Shapiro Wilk test for normality and the Levene's test for equal variances.

(a) P-value of the Levene's test for equal variances.

| | Loc - BB |
|---|---|
| Precision | 0.94 |
| Recall | 0.91 |
| F1 | 0.93 |

(b) We report different performance metrics and run a Shapiro Wilk test for normality for every metric.

| | Loc | BB |
|---|---|---|
| Precision | 88.51 $\pm 7.22$ | 89.22 $\pm 6.81$ |
| Shapiro Wilk | 0.0251 | 0.1068 |
| Recall | 88.48 $\pm 4.7$ | 86.44 $\pm 4.91$ |
| Shapiro Wilk | 0.7427 | 0.1568 |
| F1 | 88.19 $\pm 3.31$ | 87.51 $\pm 3.32$ |
| Shapiro Wilk | 0.1739 | 0.1572 |

(c) We did not find significant differences, indicating that there is no expected bias due to the annotation tools.

| | Loc - BB |
|---|---|
| Precision | 0.6858 |
| Recall | 0.0864 |
| F1 | 0.3127 |

Table 17: We directly compare all annotations by reporting precision, recall and F1 metrics. We report the p-value of the conducted statistical tests for all metrics. Significant differences are marked in bold. The p-value is the result of a paired t-test or the Wilcoxon signed-rank test, depending on the outcome of the Shapiro Wilk test for normality and the Levene's test for equal variances.

(a) P-value of the Levene's test for equal variances.

| | Bin - Loc - BB |
|---|---|
| Precision | 0.0472 |
| Recall | 0.2442 |
| F1 | 0.0308 |

(b) We report different performance metrics and run a Shapiro Wilk test for normality for every metric.

| | Bin | Loc | BB |
|---|---|---|---|
| Precision | 98.83 $\pm 1.26$ | 88.51 $\pm 7.22$ | 71.12 $\pm 12.17$ |
| Shapiro Wilk | 0.0411 | 0.0251 | 0.2794 |
| Recall | 95.24 $\pm 5.13$ | 88.48 $\pm 4.7$ | 69.04 $\pm 11.24$ |
| Shapiro Wilk | 0.1613 | 0.7427 | 0.4118 |
| F1 | 96.9 $\pm 2.39$ | 88.19 $\pm 3.31$ | 69.85 $\pm 11.3$ |
| Shapiro Wilk | 0.2381 | 0.1572 | 0.5273 |

(c) We found significant differences between all three tasks indicating a correlation between annotation error and task complexity.

| | Bin - Loc | Bin - BB | Loc - BB |
|---|---|---|---|
| Precision | **0.0312** | **0.0312** | **0.0312** |
| Recall | 0.1253 | **0.0025** | **0.0064** |
| F1 | **0.0040** | **0.0312** | **0.0312** |

Table 18: We compare the mental load and task difficulty over all tasks with a Likert scale from one (low) to nine (high). We report the p-value of the conducted statistical tests for all metrics. Significant differences are marked in bold. The p-value is the result of a paired t-test or the Wilcoxon signed-rank test, depending on the outcome of the Shapiro Wilk test for normality and the Levene's test for equal variances.

(a) P-value of the Levene's test for equal variances.

|  | Bin - Loc - BB |
| --- | --- |
| Mental Load | 0.5772 |
| Difficulty | 0.0645 |

(b) We report the median of the nine point likert scale and run a Shapiro Wilk test for normality for every metric.

|  |  | Bin | Loc | BB |
| --- | --- | --- | --- | --- |
| Mental Load | Median | 2.0 | 4.5 | 6.0 |
|  | Shapiro Wilk | 0.1670 | 0.2517 | 0.1492 |
| Difficulty | Median | 1.0 | 3.0 | 3.5 |
|  | Shapiro Wilk | 0.0063 | 0.3515 | 0.9605 |

(c) We found significant differences between all three tasks regarding the reported mental load. Additionally, we found significant differences in the difficulty of annotating bounding boxes or binary labels.

|  | Bin - Loc | Bin - BB | Loc - BB |
| --- | --- | --- | --- |
| Mental Load | **0.0117** | **0.0047** | **0.0335** |
| Difficulty | 0.0655 | **0.0421** | 0.2031 |

# D    QUALITATIVE RESULTS

## D.1    ROBUSTNESS OF OPTIMIZATION APPROACH

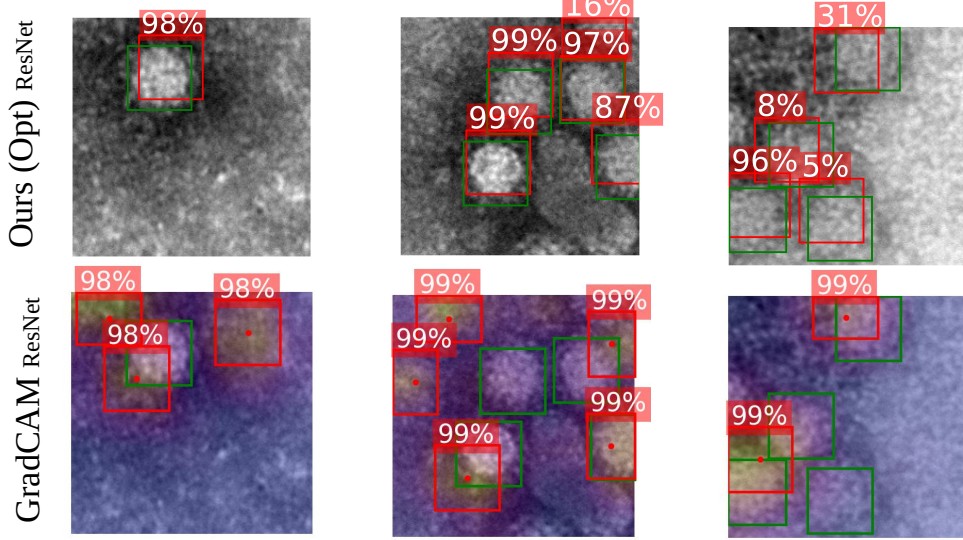

Figure 9: Even though there is a clear bias visible towards the boarder of the virus (which is most likely due to the imaging modality of negative stain TEM), our approach is able to converge to suitable positions of the virus based on the introduced optimization strategy. The GradCAM approach, which was applied to the same classifier, on the other hand, is not able to produce well fitting bounding boxes.

## D.2 COMPARISON TO WEAKLY SUPERVISED APPROACHES

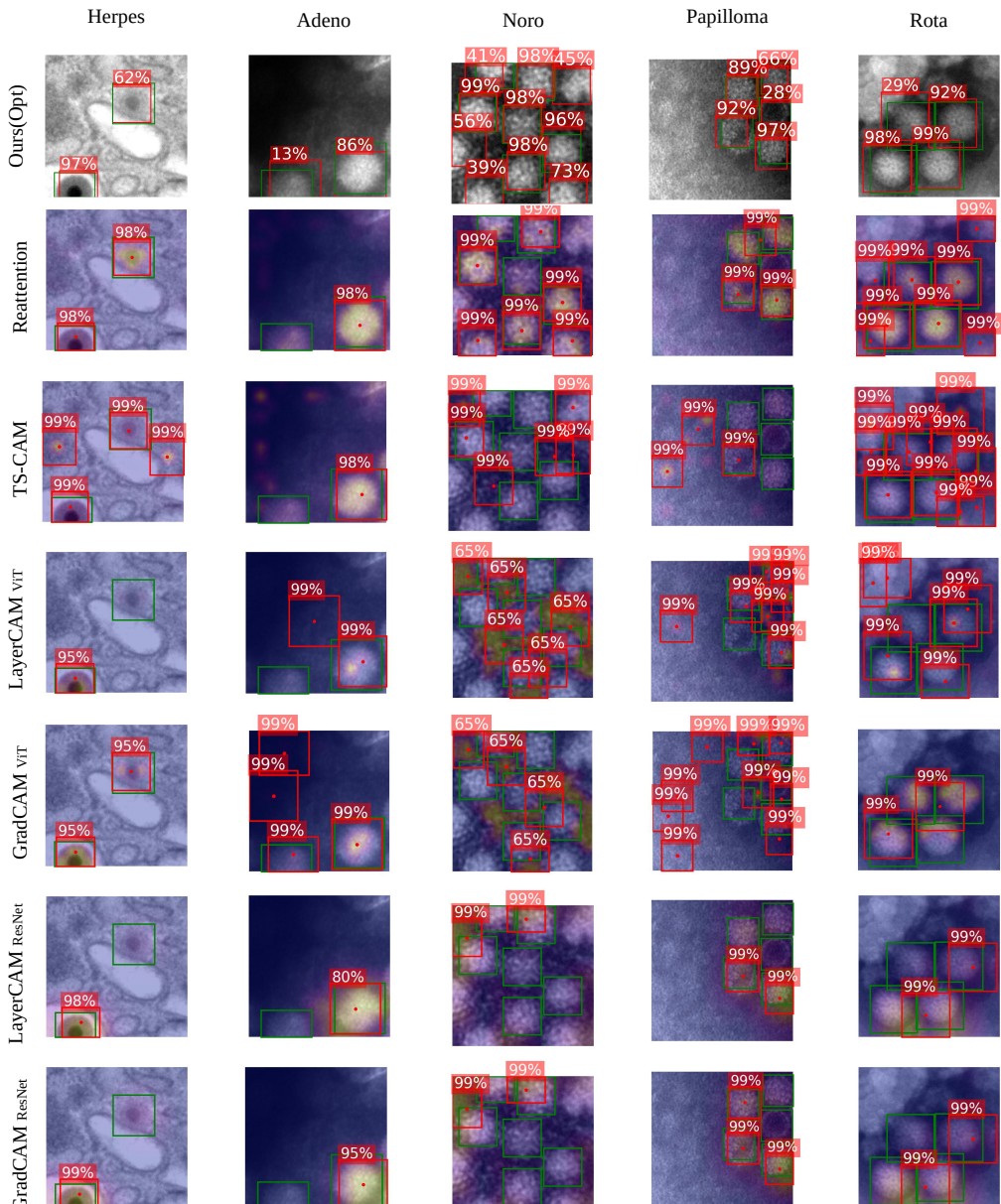

Figure 10: We compare our pseudolabel generation (Ours(Opt)) to other weakly supervised approaches and show qualitative results for all viruses.

## D.3 VIRUS SIZE FOR SAM AND CUTLER

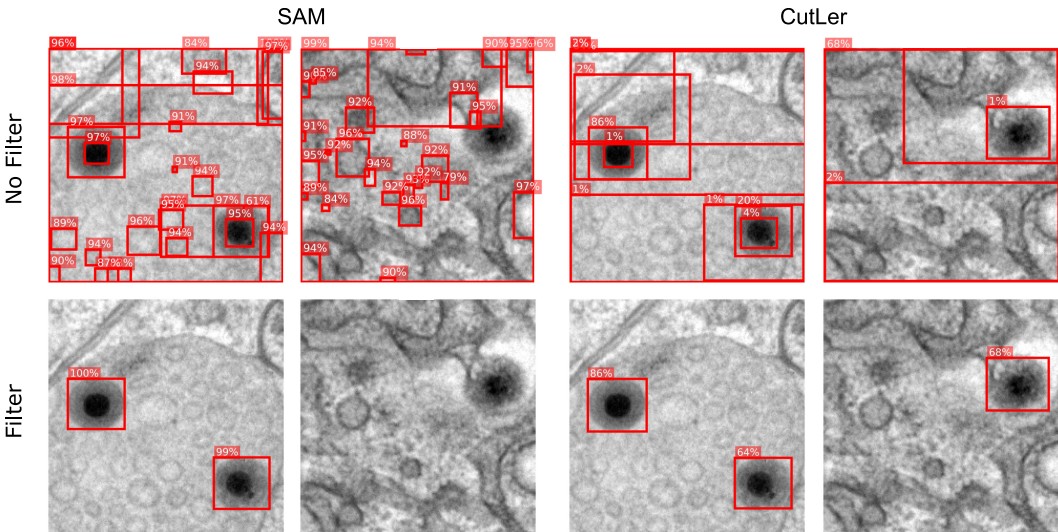

Figure 11: For a fair comparison to our approach we also inform the bounding boxes generated by Segment Anything Model (SAM) and Cut and Learn (CutLER) about the known virus size. As shown above, we are hence able to reduce the amount of FPs

### D.4 COMPARISON TO SAM AND CUTLER

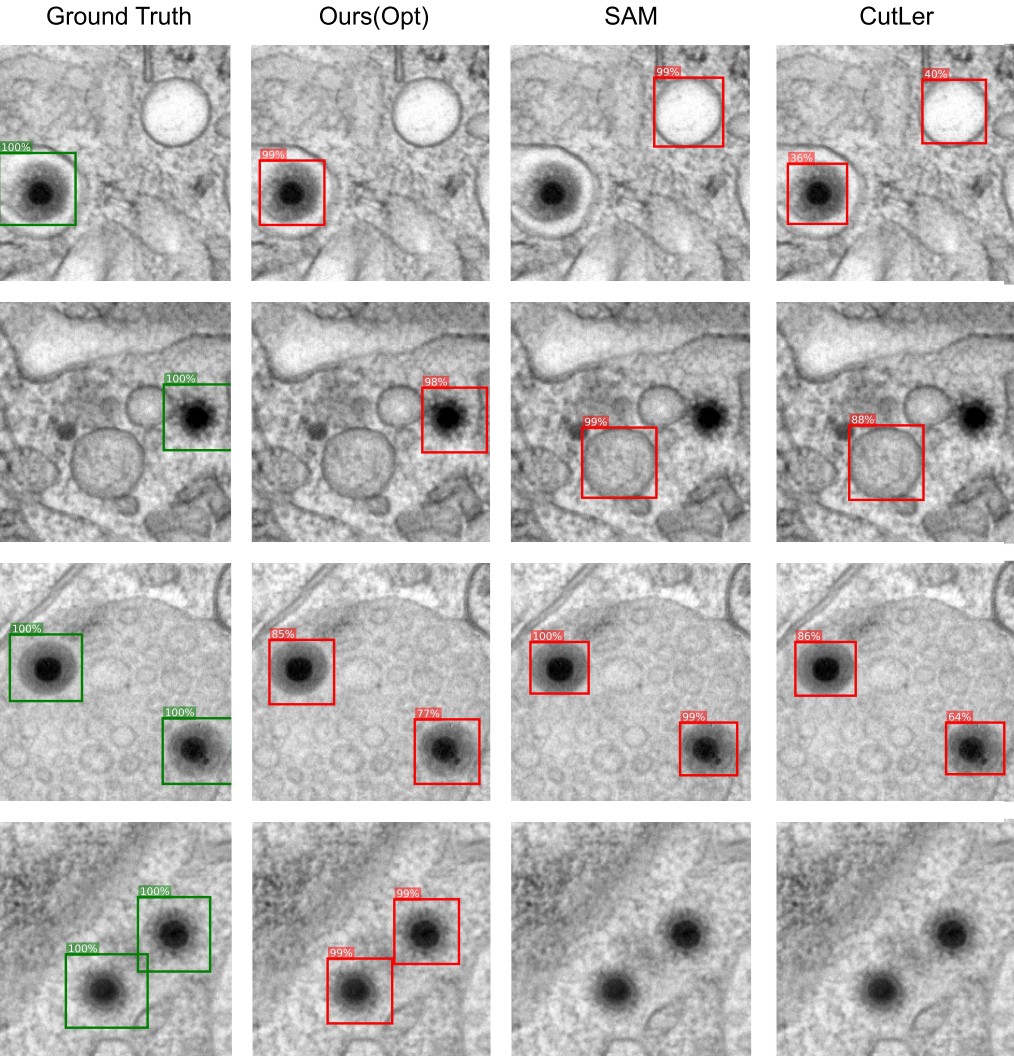

Figure 12: A qualitative comparison of the pseudolabels generated by SAM, CutLER and Ours(Opt). We found that CutLER and SAM are able to detect enveloped virus capsids comparably comparably well (row 4). We believe that this is the case due to the enclosing membrane around the capsid, that clearly outlines the object. However, failure cases of SAM and CutLER include the detection of virus-unrelated vesicles (row 1 + 2) as well as the inability to detect naked virus capsids (row 2+4).

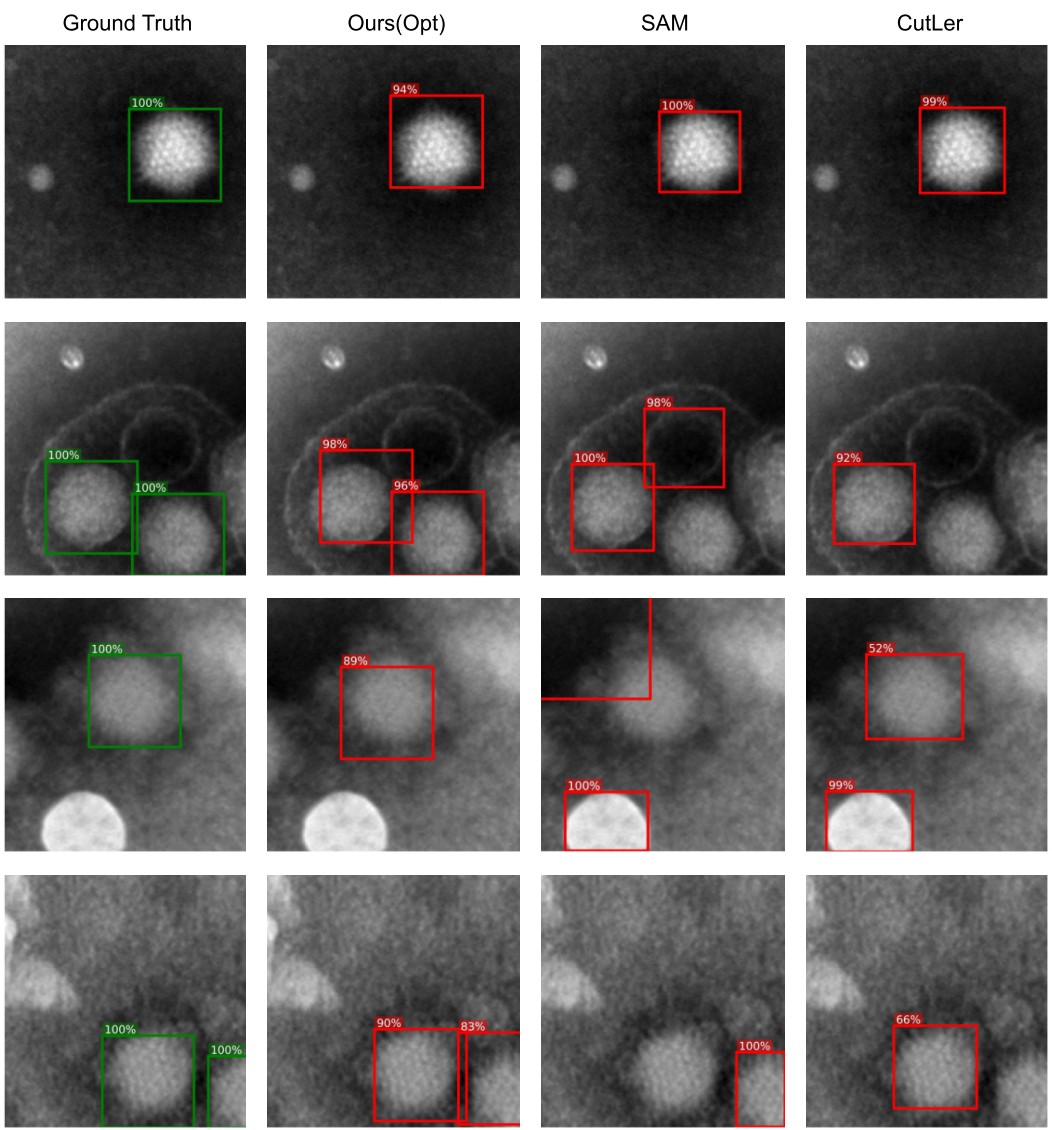

Figure 13: A qualitative comparison of the pseudolabels generated by SAM, CutLER and Ours(Opt) for the Adeno virus. While all approaches are able to perform well on high contrast virus particles (top row) SAM and CutLER fail to detect low contrast viruses. Additionally, unrelated, similar shaped, high contrast objects are also detected, due to the zero shot learning of the base model.

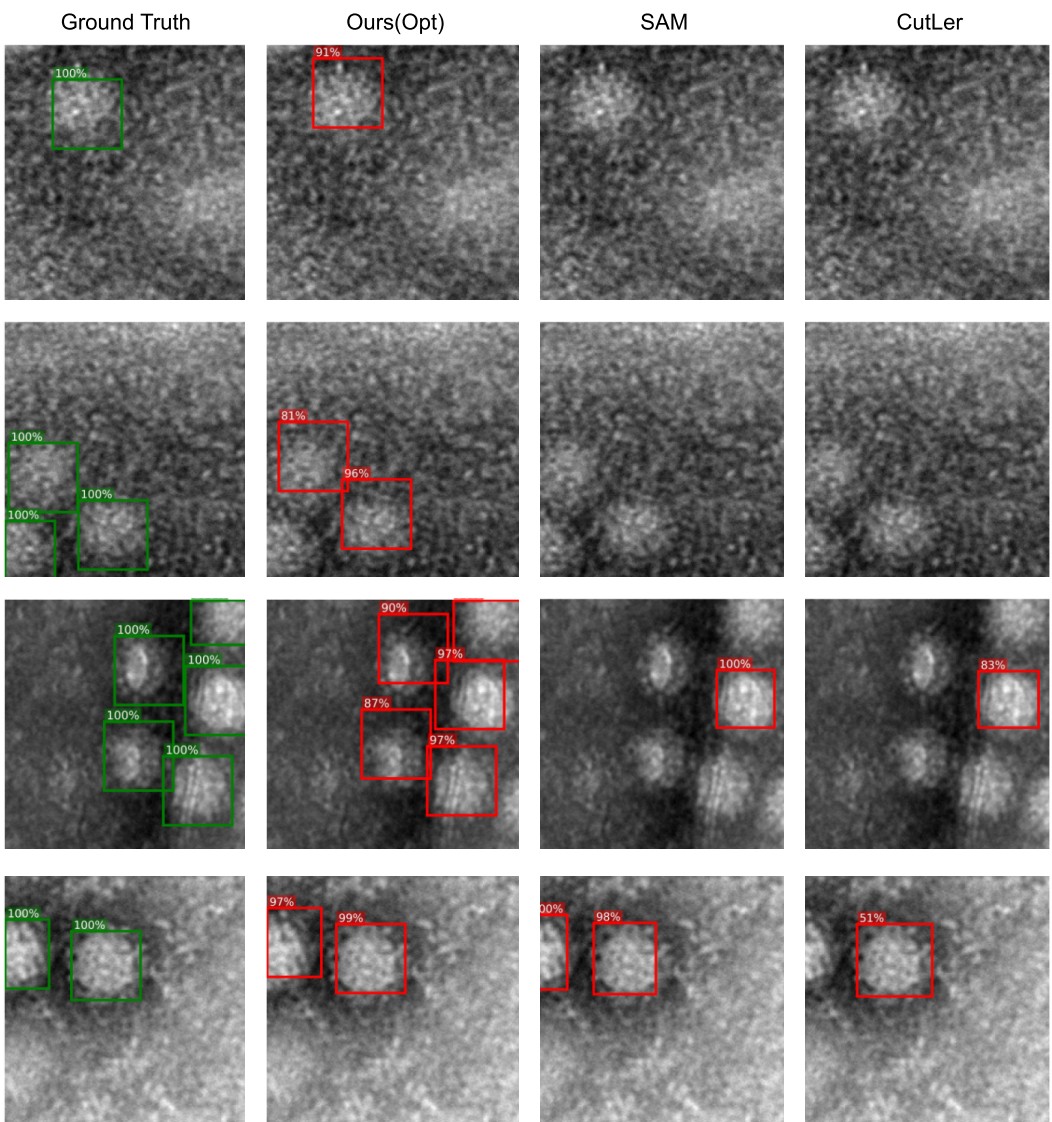

Figure 14: A qualitative comparison of the pseudolabels generated by SAM, CutLER and Ours(Opt) for the Noro virus. As the data of the Noro virus is rather noisy, we can here see the main benefits of our approach being able to detect almost not visible virus particles (top row). Still, on higher contrast virus particles, again SAM and CutLER perform comparably well.

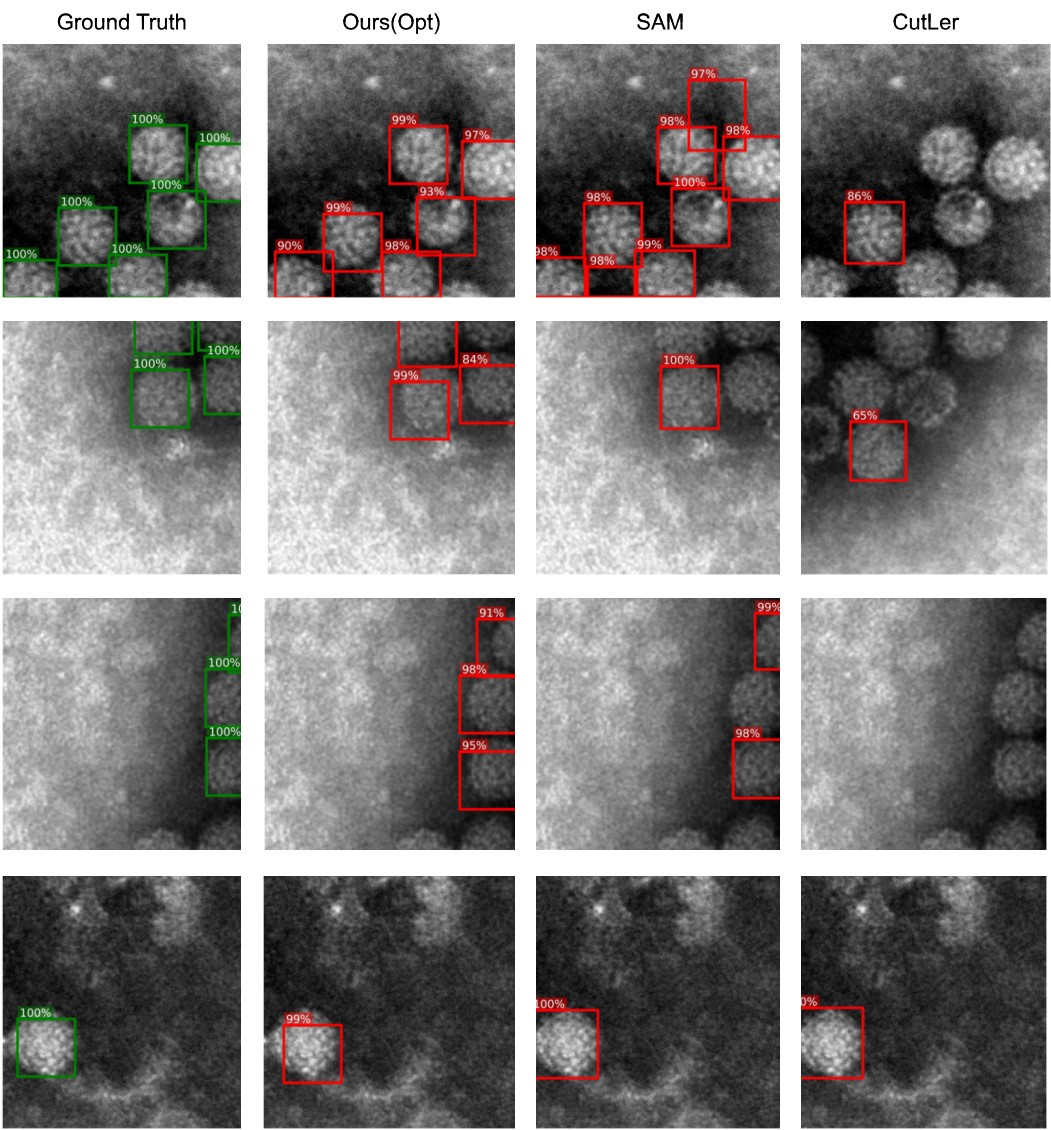

Figure 15: A qualitative comparison of the pseudolabels generated by SAM, CutLER and Ours(Opt) for the Papilloma virus. For the this virus SAM as well CutLER struggle to detect multiple objects in an image. While SAM is still able to perform well on some images, CutLER usually only detect a single instance.

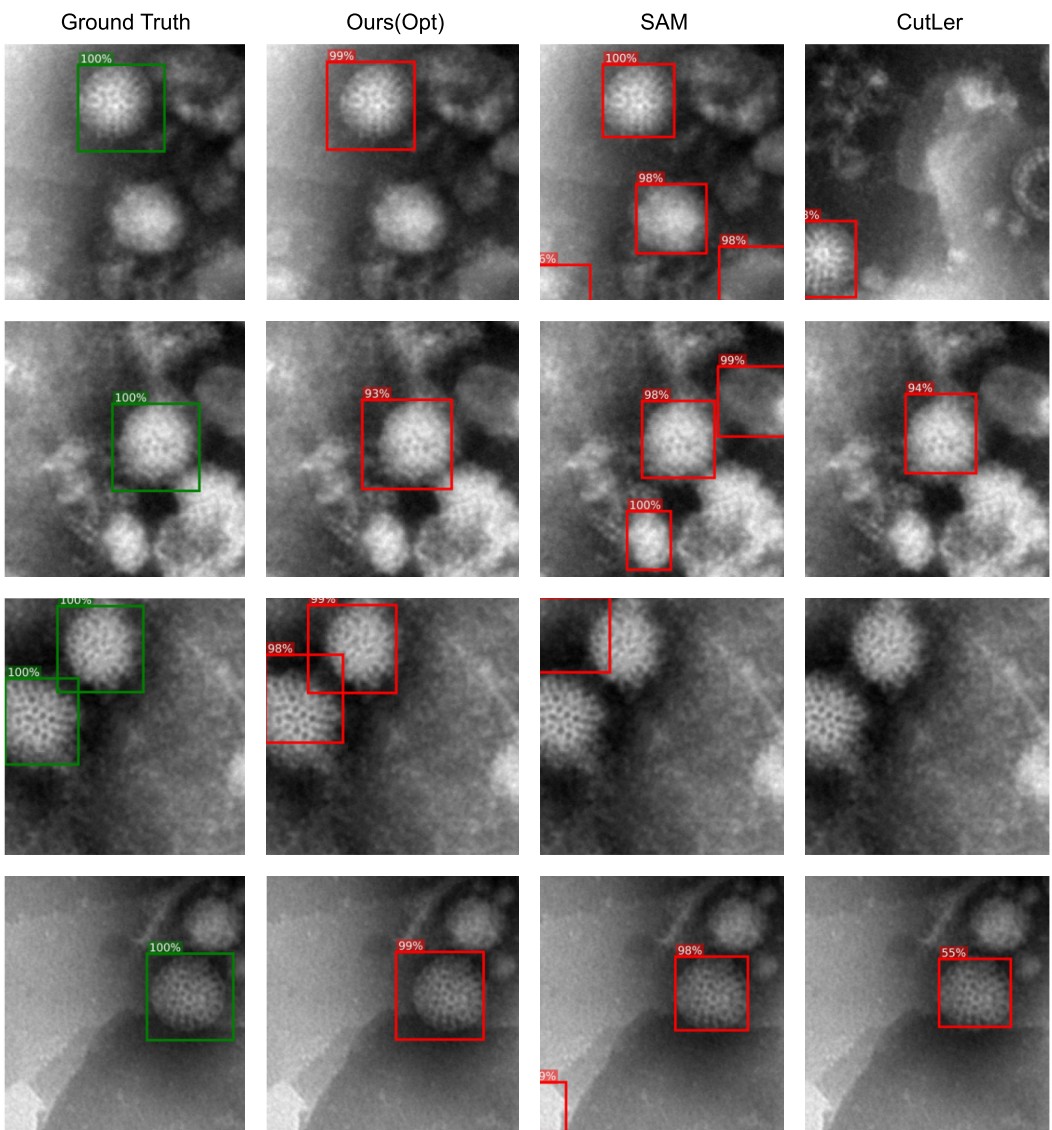

Figure 16: A qualitative comparison of the pseudolabels generated by SAM, CutLER and Ours(Opt) for the Rota virus. We can observe similar behaviours of the pseudolabels as for the other virus particles.

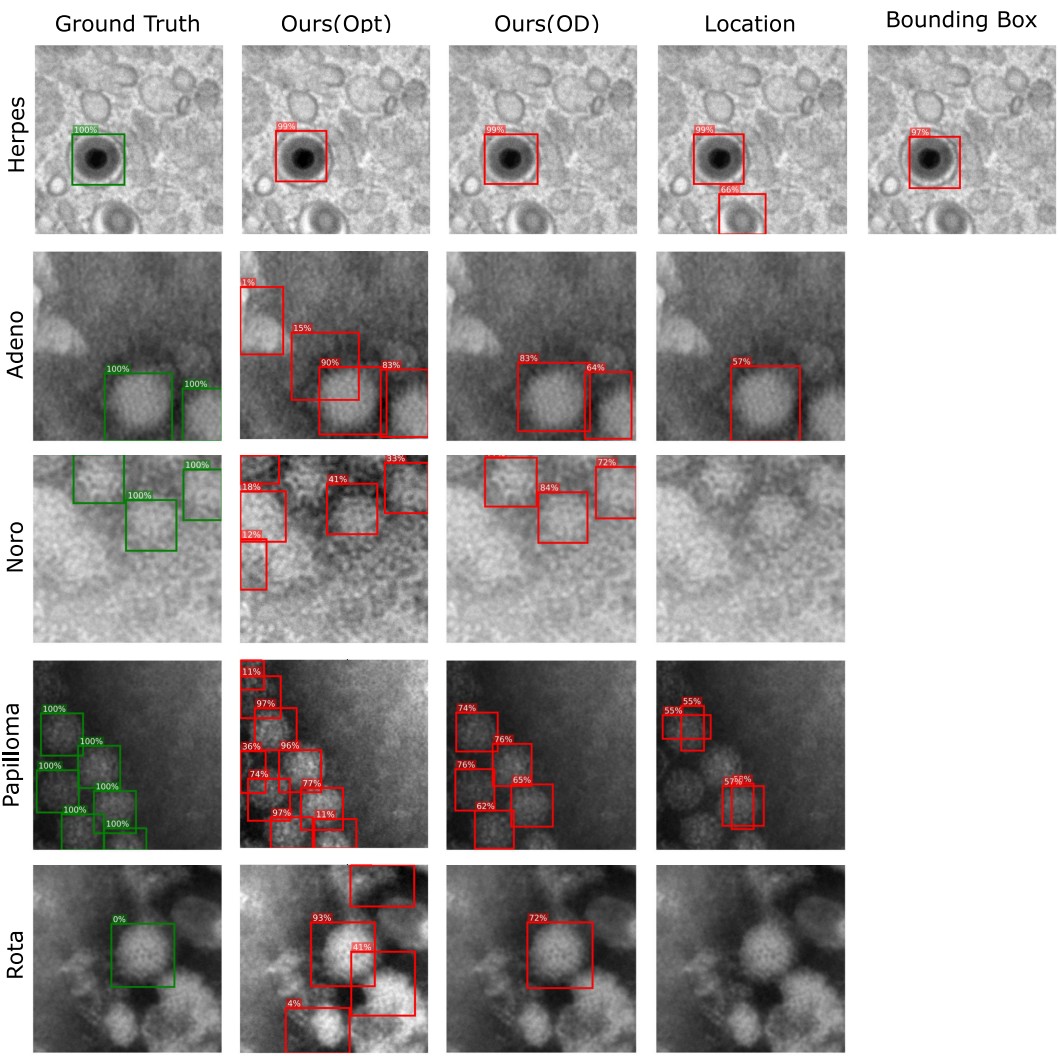

Figure 17: Qualitative results of the predictions of the Faster-RCNN, trained on bounding box labels, location labels and image level labels. Additionally, we show predictions of our weakly supervised approach Ours(Opt) directly on the test set. We found that Ours(Opt) tends to predict FP. However, these FPs are detected with low score, allowing the Faster-RCNN to learn a differentiation between FPs and TPs. This explains the superior performance of Ours(OD) in comparison with Ours(Opt).

