# OpenReview forum: "Weakly Supervised Virus Capsid Detection with Image-Level Annotations in Electron Microscopy Images"
_ICLR.cc/2024/Conference — ICLR 2024 poster_

### Official Review · Reviewer_KB1H · 2023-10-29

**Soundness:** 3 good
**Presentation:** 3 good
**Contribution:** 3 good
**Rating:** 5
**Confidence:** 4

**Summary:**

This paper addresses the challenge of expensive and time-consuming annotation requirements for training state-of-the-art object detection models. The authors propose a domain-specific weakly supervised object detection algorithm that leverages image-level annotations instead of annotated bounding boxes. By distilling the knowledge of a pre-trained model focused on virus presence/absence prediction, the proposed approach generates a set of pseudo-labels that can be used to train an object detection model effectively. The method utilizes an optimization approach with a shrinking receptive field, enabling the extraction of virus particles directly without relying on specific network architectures.

**Strengths:**

- Addressing Expensive Annotation Requirement: The paper tackles the challenge of acquiring costly bounding box annotations by proposing a weakly supervised approach that relies on image-level annotations. This significantly reduces the manual labor and time required for annotation, making the training process more efficient.
- Extensive Comparative Studies: The authors conduct comprehensive studies to evaluate the effectiveness of the proposed pseudo-labels. The results demonstrate that the generated pseudo-labels outperform other weak labeling methods and even ground truth annotations in scenarios where annotation time is limited. This indicates the superiority and practical value of the proposed approach.

**Weaknesses:**

- Limited Model Exploration. The paper primarily focuses on using Faster-RCNN with a ResNet-101 backbone as the detection model. It would be beneficial for the authors to consider exploring other models, such as DETR, to evaluate their effectiveness in the proposed approach.
- Lack of Discussion on Low Signal to Noise Ratio (SNR) in EM Images: While the authors mention that low SNR in EM images can impact the performance of methods designed for other imaging modalities, there is a lack of in-depth discussion, algorithm design, and experiments addressing how to mitigate the low SNR problem and how it specifically affects the capacity of weakly supervised object detection (WSOD) methods in the EM scenario. Further exploration and discussion of this property would be valuable to enhance the understanding and applicability of the proposed approach.

**Questions:**

In general, I find this work to be commendable. However, there are certain limitations that should be addressed for further improvement. Specifically, it is crucial to include testing with DETR to evaluate the effectiveness of transformer-based architectures. Additionally, at least providing a thorough discussion on the low SNR problem would significantly enhance the quality of the paper. If these questions are well solved, I would like happy to raise my score.

---

> ### Author Response · Authors · 2023-11-22
>
> Dear reviewer KB1H,
> we highly appreciate your thoughtful comments. In the following, we will respond to every comment.
>
> ## Limited Model Exploration
> We agree, that our model exploration is limited. We motivate our decision to use CNN-based architectures by the limited data set sizes for virus detection in EM. However, we agree that a more thorough model exploitation can give additional insights, specifically as ViTs have shown great success in weakly supervised setups. We hence include the comparison of Ours(Opt) based on the ResNet-101 and Ours(Opt) based on a ViT backbone in the updated version of our paper (appendix A.2).
> Nevertheless, we refrained from delving deeper into the exploration of alternative detection models such as DETR. Our ablation study, employing ViT backbones for Ours(Opt), suggests that DETR might face challenges and likely underperform, particularly given the constraints posed by the relatively small dataset sizes. However, we acknowledge that leveraging different detection models has the potential to enhance the performance of all methods, including our proposed approach (Ours(OD)).
>
> ## Lack of Discussion on Low Signal to Noise Ratio (SNR) in EM Images
> We agree that the low SNR in EM images can lead to the current state-of-the-art methods not performing well on EM images. Additionally, there are more challenges that are being posed by the nature of virus detection in EM like the occurrence of multiple instances of the same object in one image, as well as small annotated dataset sizes.
> We hence include a more thorough discussion of the topic in relation to our updated evaluation in the paper (see section 4.4).

---

### Official Review · Reviewer_zGyY · 2023-10-29

**Soundness:** 3 good
**Presentation:** 3 good
**Contribution:** 2 fair
**Rating:** 6
**Confidence:** 5

**Summary:**

The manuscript presents a class activation map (CAM)-based weakly supervised learning method for virus particle detection in electron microscopy images. Specifically, it first uses a pre-trained classifier to obtain an initial position of a virus using GradCAM (Selvaraju et al. 2017), and then iteratively refines the position with a Gaussian mask with a dynamic standard deviation. It repeats this process for each virus until all the viruses are detected in the input image. The proposed method is evaluated on 5 electron microscopy image datasets, and the experimental results are promising.

**Strengths:**

1. The paper introduces a simple yet promising method for weakly supervised object detection. Meanwhile, it conducts extensive ablation studies to show that the proposed weakly supervised method can outperform other more fine-grained annotation-based approaches (e.g., bounding box and point annotations), given a certain time budget.

2. The paper designs a specific user study to demonstrate the effectiveness and efficiency of the proposed method.

**Weaknesses:**

1. In the experiments, the proposed method is not compared with other state-of-the-art weakly supervised learning methods, such as Zeng et al. 2019, Wei et al. 2022, and Lu et al. 2020. In addition, it is not compared with other CAM-based weakly supervised object detection methods in the experiments, such as Xu et al. 2022. Without a comparison with recent state of the art, it is difficult to determine the superiority of the proposed methods over other approaches.

2. The method requires the object size to be known in advance. This needs additional effort to estimate the size of target objects before applying the method. It would be helpful to provide an in-depth discussion on this design (probably also including the effects of using different estimated object sizes).

3. It seems that the proposed method needs to repeat the optimization process (i.e., solving Equation (2)) for each virus. The time cost may be high if there is a large number of viruses in the input image.

4. The method is evaluated on only virus detection in electron microscopy images, where viruses do not overlap. Thus, the method may not generalize to object detection (e.g., cell or nuclei detection) in other microscopy imaging modalities, such as hematoxylin and eosin (H&E) or immunohistochemistry (IHC) stained brightfield microscopy images, and fluorescence mages, which often have touching or overlapping cells or nuclei. In addition, the repeated optimization for each object would be expensive for H&E or IHC images that typically have thousands of or even more cells/nuclei.

**Questions:**

1. The proposed method is based on the GradCAM method. What if the GradCAM does not provide good initializations or even wrong saliency maps? What are the effects of inaccurate saliency map creation on the quality of the pseudo-labels generated by the proposed method?

2. During the optimization process of the proposed method, i.e., solving Equation (2), is the classifier C fixed and not updated? If so, is the optimization of Equation (2) simply to find the position that has the highest value in the prediction map C(I * M(p_t)) at each time step? But if not, what algorithm is used to optimize Equation (2)?

3. During the postprocessing, the method uses non-maximum suppression to eliminate virus particles that have low detection scores. Are the detection scores of virus particles obtained from the initial CAM map or the prediction map from the Gaussian-filtered input, C(I * M(p_t))?

---

> ### Author Response · Authors · 2023-11-22
>
> Dear Reviewer zGyY,
> We highly appreciate your comments and will discuss them in detail in the following. We are looking forward to further discussions.

---

> ### Author Response · Authors · 2023-11-22
> **Comparison to other WSOL methods**
>
> We acknowledge this concern and regret that certain methods, such as [1] and [2], lacked publicly available training code. Other methods like GC-Net [3] lack the ability to detect more than one object, which is why we chose not to include it in the comparisons. We opted to include alternative state-of-the-art methods — specifically, LayerCAM [4], TS-CAM [5], and Reattention [6]. We further included GradCAM [7], as we make use of this method as an initialization schema. As ViTs have shown great success in current state of the art WSOL methods, we include the ViT as well as ResNet backbones for the application of GradCAM and LayerCAM.
> In the interest of a more equitable comparison, we opted to integrate the known virus size into the existing methods. Through an extensive ablation study (appendix A.5), we systematically determined the most suitable approach for informing these methods (section 4.4).
>
> | Method                        | Herpes                | Adeno                   | Noro                 | Papilloma                      | Rota                 |
> |-------------------------------|--------------------------|--------------------------|--------------------------|--------------------------|--------------------------|
> | $\mathrm{GradCAM}$ ResNet      | 78.79 $\pm$ 2.04        | 19.17 $\pm$ 0.78        | 05.54 $\pm$ 2.99        | 11.57 $\pm$ 4.17        | 31.78 $\pm$ 21.58       |
> | $\mathrm{LayerCAM}$ ResNet     | 78.44 $\pm$ 2.73        | 16.48 $\pm$ 9.34        | 05.04 $\pm$ 1.91        | 10.87 $\pm$ 5.33        | 31.22 $\pm$ 20.07       |
> | $\mathrm{GradCAM}$ ViT        | 61.87 $\pm$ 11.87       | 08.00 $\pm$ 2.12        | 19.31 $\pm$ 13.64       | 04.03 $\pm$ 4.52        | 13.12 $\pm$ 7.37        |
> | $\mathrm{LayerCAM}$ ViT       | 68.33 $\pm$ 6.59        | 09.18 $\pm$ 5.64        | 10.82 $\pm$ 11.78       | 17.41 $\pm$ 11.33       | 09.74 $\pm$ 2.42        |
> | $\mathrm{TS-CAM}$             | 32.06 $\pm$ 1.02        | 39.25 $\pm$ 4.13        | 14.64 $\pm$ 4.66        | 07.11 $\pm$ 3.85        | 43.53 $\pm$ 3.93        |
> | $\mathrm{Reattention}$        | 68.85 $\pm$ 0.62        | **58.49** $\pm$ 2.22        | 55.09 $\pm$ 8.92        | 35.60 $\pm$ 13.01       | 59.05 $\pm$ 11.40       |
> | **$\mathrm{Ours (Opt)}$** | 86.98 $\pm$ 1.92  | 47.85 $\pm$ 11.82 | 54.65 $\pm$ 4.94  | 70.02 $\pm$ 2.85  | 71.73 $\pm$ 3.51 |
> | **$\mathrm{Ours (OD)}$**  | **91.20** $\pm$ 0.24 | **58.28** $\pm$ 5.91 | **74.32** $\pm$ 1.18 | **78.33** $\pm$ 2.40 | **78.34** $\pm$ 2.15 |
>
> As shown in the table above, we found that other weakly supervised methods fail to reliably detect virus particles in EM images, eventhough we also incorperate the virus size in existing methods. Our methods usually outperforms all others by a large margin, except in the adeno virus, where we are en par with Reattention. Please find a detailed discussion in the paper (section 4.4, appendix A.5).

---

> ### Author Response · Authors · 2023-11-22
> **Known object size**
>
> We regret we were not able to communicate it properly in the paper. Usually, the virus size can be found in literature. Moreover, as the standard deviation of virus particles is small, the size can easily be approximated by measuring only a few instances. We will include a discussion in the updated version of our paper in appendix A.4.
> We also agree, that the correct approximation of the objects size is important for our method in order to work well. To communicate this, we added an additional experiment in appendix A.4 about the influence of approximation error in the updated version of our paper. For a quick overview please see the provided table:
>
> | Error               | $0\%$                   | $10\%$                  | $20\%$                  | $30\%$                  |
> |---------------------|-------------------------|-------------------------|-------------------------|-------------------------|
> | $\mathrm{Ours (Opt)}$ | **86.98** $\pm$ 1.92   | 82.38 $\pm$ 02.34       | 79.14 $\pm$ 03.10       | 67.31 $\pm$ 02.03       |
>
>
> As expected, we found that the performance reduces with increased error in the size approximation.

---

> ### Author Response · Authors · 2023-11-22
> **Repetition for multiple objects**
>
> We agree that the computation time increases with the amount of visible virus particles in the image.
> This is however a specific design choice for dealing with the non-object centered nature of EM images.
> Our evaluation shows that current state of the art methods lack to reliably detect multiple instances of the same object as they only have been trained on image level labels, and hence are able to only focus on the most prominent virus in the image (see appendix C.2). Our approach is able to side step this limitation to be applicable in the context of EM.

---

> ### Author Response · Authors · 2023-11-22
> **Generalization to other microscopy modalities**
>
> In general, the detection of overlapping instances of the same object in a weakly supervised setup is no trivial task and remains a major challenge. We would like to emphasize that we specifically designed an approach for the detection of virus particles in EM images. In our extended evaluation we can show that current state of the art approaches fail to reliably detect virus particles in EM, which highlights the need of a specific method.
> Addressing the broader challenge of detecting overlapping instances of the same object within a weakly supervised setup is inherently complex and represents a substantial hurdle in general weakly supervised setups. Nonetheless, we maintain our confidence in the adaptability of our approach to diverse microscopy modalities. Qualitative results presented in the appendix (section C) illustrate instances where virus particles form clusters and exhibit touching behavior. Notably, our current method focuses exclusively on non-overlapping viruses, employing non-maximum suppression with a strict threshold to mitigate overlap.
> We believe that relaxing this threshold would likely enable our method to detect overlapping instances. However, a thorough evaluation and extension of our method in this regard are reserved for future research.

---

> ### Author Response · Authors · 2023-11-22
> **GradCAM initialization**
>
> In the Appendix A.1.1 we investigate different methods for initialization and the power of the optimization process. The experiment shows, that even with non-optimal initialization (like random initialization) the optimization process is able to converge to an optimum, when enough iterations are used. In our extended evaluation, we found that GradCAM based WSOL failed to reliably detect the virus capsids in negative stain images (see appendix C.1), as the classifier was focusing on the boarder of the virus. However, the inclusion of our optimization procedure enables a more robust detection.

---

> ### Author Response · Authors · 2023-11-22
> **Optimization of Equation 2**
>
> Yes, the classifier is fixed during the optimization. We will emphasize this more in the updated version of our paper.

---

> ### Author Response · Authors · 2023-11-22
> **Computation of detection scores**
>
> We kindly refer to Section 3.5 in our paper. Here we explain, that the detection score is computed individually for each detected virus in a postprocessing step. To do so, we mask all other detected virus particles by a circular disk with radius of the known virus size and derive the score of the remaining virus by forwarding the masked image to the pretrained classifier.

---

> ### Author Response · Authors · 2023-11-22
> **References**
>
> [1] Xu, J., Hou, J., Zhang, Y., Feng, R., Zhao, R.W., Zhang, T., Lu, X. and Gao, S., 2022. Cream: Weakly supervised object localization via class re-activation mapping. In Proceedings of the IEEE/CVF Conference on Computer Vision and Pattern Recognition (pp. 9437-9446).
>
> [2] Wei, J., Wang, S., Zhou, S.K., Cui, S. and Li, Z., 2022, October. Weakly supervised object localization through inter-class feature similarity and intra-class appearance consistency. In European Conference on Computer Vision (pp. 195-210). Cham: Springer Nature Switzerland.
>
> [3] Weizeng Lu, Xi Jia, Weicheng Xie, Linlin Shen, Yicong Zhou, and Jinming Duan. Geometry constrained weakly supervised object localization. In European Conference on Computer Vision (ECCV), 2020
>
> [4] Jiang, Peng-Tao, et al. "Layercam: Exploring hierarchical class activation maps for localization." IEEE Transactions on Image Processing 30 (2021): 5875-5888.
>
> [5] Gao, Wei, et al. "Ts-cam: Token semantic coupled attention map for weakly supervised object localization." Proceedings of the IEEE/CVF International Conference on Computer Vision. 2021.
>
> [6] Su, Hui, et al. "Re-attention transformer for weakly supervised object localization." arXiv preprint arXiv:2208.01838 (2022).
>
> [7] Selvaraju, Ramprasaath R., et al. "Grad-cam: Visual explanations from deep networks via gradient-based localization." Proceedings of the IEEE international conference on computer vision. 2017.

---

### Official Review · Reviewer_6zGs · 2023-11-01

**Soundness:** 4 excellent
**Presentation:** 3 good
**Contribution:** 2 fair
**Rating:** 5
**Confidence:** 4

**Summary:**

This work presents a method for weakly-supervised object detection (WSOL) of virus capsids in EM images, which can be used for rapid curation of bounding boxes. The overview of this method (presented in Figure 1) uses an iterative process in which: 1) Grad-CAM from a pretrained encoder is use to output saliency maps of the highest-scoring virus location, 2)  gradient descent is used to optimize the location of the virus, 3) virus is masked out using known information about the virus size. This process repeats until all viruses are removed (referenced as Ours (Opt)), with the bounding boxes created using this process usable for developing weakly-supervised object detectors (Ours (OD)). Comparisons against human annotators (weakly-supervised binary annotation, location, bounding box) and self-supervised detectors were performed, with comparison against human annotators (with and without time constraints) also performed.

**Strengths:**

Overall, this work presents a very unique methodology and study design for curating bounding boxes in EM images. A contribution not emphasized in this work is the simplicity of the method, using a very intuitive heuristic that outperforms current unsupervised, deep learning-based detectors such as SAM and CUTLER. Though specific to EM, I believe the uniqueness and simplicity of this work would still be of interest to the computer vision community. The related work section is all comprehensive, and the authors of this work reference related works in weakly-supervised and self-supervised object detection very well.

**Weaknesses:**

- Though the related work section provides a comprehensive overview of current progress in WSOL methods, was there a reason why this work does not compare against other WSOL methods such as Xu et al. [1] (CREAM), Wei et al. [2] (ISIC), and other more recent works such as LOCATE [3] and GenPromp [4]? Though specific to EM, many other works in the WSOL domain can also be readily adapted.
- In addition to lack of comparisons, one of the main limitations of this work that may prevent broader interest in the ICLR community is that the proposed method is too specific to EM and is not evaluated on diverse tasks. Though EM is unique compared to natural images which are generally more object-centric, other modalities such as histopathology and multiplexed imaging share similar characteristics (as noted in [5]), with the image scale is objective with units per pixel being fixed. Specifically, the contributions of this work would be strengthened if shown that a simpler heuristic can also be created for other imaging domains.
- Following other works which have found pretrained Vision Transformers (ViTs) to be strong in WSOL [4,5,6], was there as a reason why a ResNet-101 was used for classification instead of a ViT? Moreover, was the DINO-ViT used in CUTLER trained using EM images, or was it using a pretrained checkpoint from ImageNet? As ViTs have been also found to have natural fit for microscopy images [5], it would be interesting to explore how the a DINO-ViT for EM images would: 1) improve the WSOL results of this work, and 2) improve the CUTLER baseline reported in this work.
- Though this work is well-written, it was difficult to understand the training dataset and the downstream dataset for evaluation and annotator labeling. Though described in text, including a table with the distribution of labels for train and test may be simpler to communicate.

Overall, I found this work to have a unique contribution computer vision in proposing simpler techniques that outperform more complicated solutions that are more complex to train and underperform in domain-specific areas. At the same time, I feel that the scope of the work is too narrow for ICLR, and lacks comparisons to other WSOL works. I still feel that this work has many strengths, and believe that it would make a timely contribution in conferences specific to computer vision such as CVPR and ECCV where approaches for solving domain-specific challenges would be more broadly appreciated.

1. Xu, J., Hou, J., Zhang, Y., Feng, R., Zhao, R.W., Zhang, T., Lu, X. and Gao, S., 2022. Cream: Weakly supervised object localization via class re-activation mapping. In Proceedings of the IEEE/CVF Conference on Computer Vision and Pattern Recognition (pp. 9437-9446).
2. Wei, J., Wang, S., Zhou, S.K., Cui, S. and Li, Z., 2022, October. Weakly supervised object localization through inter-class feature similarity and intra-class appearance consistency. In European Conference on Computer Vision (pp. 195-210). Cham: Springer Nature Switzerland.
3. Li, G., Jampani, V., Sun, D. and Sevilla-Lara, L., 2023. LOCATE: Localize and Transfer Object Parts for Weakly Supervised Affordance Grounding. In Proceedings of the IEEE/CVF Conference on Computer Vision and Pattern Recognition (pp. 10922-10931).
4. Zhao, Y., Ye, Q., Wu, W., Shen, C. and Wan, F., 2023. Generative prompt model for weakly supervised object localization. In Proceedings of the IEEE/CVF International Conference on Computer Vision (pp. 6351-6361).
5. Chen, R.J., Chen, C., Li, Y., Chen, T.Y., Trister, A.D., Krishnan, R.G. and Mahmood, F., 2022. Scaling vision transformers to gigapixel images via hierarchical self-supervised learning. In Proceedings of the IEEE/CVF Conference on Computer Vision and Pattern Recognition (pp. 16144-16155).
6. Murtaza, S., Belharbi, S., Pedersoli, M., Sarraf, A. and Granger, E., 2023. Discriminative sampling of proposals in self-supervised transformers for weakly supervised object localization. In Proceedings of the IEEE/CVF Winter Conference on Applications of Computer Vision (pp. 155-165).

**Questions:**

See above.

---

> ### Author Response · Authors · 2023-11-22
>
> Dear reviewer 6zGs,
> we highly appreciate your thoughtful comments. In the following, we will respond to every comment. We are looking forward to further discussions.
>
> ## Comparison to other WSOL methods
> We acknowledge this concern and regret that certain methods, such as [1] and [2], lacked publicly available training code. Other methods like GC-Net [3] lack the ability to detect more than one object, which is why we chose not to include it in the comparisons. We opted to include alternative state-of-the-art methods — specifically, LayerCAM [4], TS-CAM [5], and Reattention [6]. We further included GradCAM [7], as we make use of this method as an initialization schema. As ViTs have shown great success in current state of the art WSOL methods, we include the ViT as well as ResNet backbones for the application of GradCAM and LayerCAM.
> In the interest of a more equitable comparison, we opted to integrate the known virus size into the existing methods. Through an extensive ablation study (appendix A.5), we systematically determined the most suitable approach for informing these methods (section 4.4).
>
> | Method                        | Herpes                | Adeno                   | Noro                 | Papilloma                      | Rota                 |
> |-------------------------------|--------------------------|--------------------------|--------------------------|--------------------------|--------------------------|
> | $\mathrm{GradCAM}$ ResNet      | 78.79 $\pm$ 2.04        | 19.17 $\pm$ 0.78        | 05.54 $\pm$ 2.99        | 11.57 $\pm$ 4.17        | 31.78 $\pm$ 21.58       |
> | $\mathrm{LayerCAM}$ ResNet     | 78.44 $\pm$ 2.73        | 16.48 $\pm$ 9.34        | 05.04 $\pm$ 1.91        | 10.87 $\pm$ 5.33        | 31.22 $\pm$ 20.07       |
> | $\mathrm{GradCAM}$ ViT        | 61.87 $\pm$ 11.87       | 08.00 $\pm$ 2.12        | 19.31 $\pm$ 13.64       | 04.03 $\pm$ 4.52        | 13.12 $\pm$ 7.37        |
> | $\mathrm{LayerCAM}$ ViT       | 68.33 $\pm$ 6.59        | 09.18 $\pm$ 5.64        | 10.82 $\pm$ 11.78       | 17.41 $\pm$ 11.33       | 09.74 $\pm$ 2.42        |
> | $\mathrm{TS-CAM}$             | 32.06 $\pm$ 1.02        | 39.25 $\pm$ 4.13        | 14.64 $\pm$ 4.66        | 07.11 $\pm$ 3.85        | 43.53 $\pm$ 3.93        |
> | $\mathrm{Reattention}$        | 68.85 $\pm$ 0.62        | **58.49** $\pm$ 2.22        | 55.09 $\pm$ 8.92        | 35.60 $\pm$ 13.01       | 59.05 $\pm$ 11.40       |
> | **$\mathrm{Ours (Opt)}$** | 86.98 $\pm$ 1.92  | 47.85 $\pm$ 11.82 | 54.65 $\pm$ 4.94  | 70.02 $\pm$ 2.85  | 71.73 $\pm$ 3.51 |
> | **$\mathrm{Ours (OD)}$**  | **91.20** $\pm$ 0.24 | **58.28** $\pm$ 5.91 | **74.32** $\pm$ 1.18 | **78.33** $\pm$ 2.40 | **78.34** $\pm$ 2.15 |
>
> As shown in the table above, we found that other weakly supervised methods fail to reliably detect virus particles in EM images, eventhough we also incorperate the virus size in existing methods and dataset sizes are the same. Our methods usually outperforms all others by a large margin, except in the adeno virus, where we are en par with Reattention. Please find a detailed discussion in the paper (section 4.4, appendix A.5).

---

> ### Author Response · Authors · 2023-11-22
> **Limited applicability of the method**
>
> Our proposed method does not aim to reach a large range of applications and object types, but it is designed for the detection of particles in electron microscopy images. In our extensive evaluation, we are able to show that the proposed method is able to outperform current state-of-the-art methods, even though we also include the known virus size in existing approaches. We believe that this is most likely due to the fact, that current state-of-the-art methods highly benefit from large dataset sizes, object-centric datasets, and high SNR. These large amounts of annotated data are not available for virus detection in EM. Additionally, EM images aren't object-centric but usually contain multiple instances of the same object in one micrograph. Finally, low SNR can hinder the performance of these methods in EM.
> This finding underlines the need for methods that are specifically designed to work well for virus detection in EM.
> Despite this, we believe that the proposed method holds significance for the broader ICLR community. Firstly, its simplicity renders it accessible and intuitive, making it potentially valuable for a wide audience. Secondly, its adaptability to a broader spectrum of applications is noteworthy. By replacing the circle with ellipses and incorporating size optimization, as suggested by [3], the approach demonstrates versatility.
> While a comprehensive evaluation of these adaptations is beyond the scope of this paper, we recognize their potential impact and plan to delve into a detailed assessment in future work.

---

> ### Author Response · Authors · 2023-11-22
> **ViTs in WSOL**
>
> We agree that ViTs have shown great success in WSOL. However, due to the small data set sizes in virus detection for EM, we decided to use basic CNN architectures, as CNNs often exhibit strong data efficiency, particularly when dealing with relatively small datasets.
> We still agree that the use of ViT as a backbone is an interesting ablation, which is why we included it in the updated version of our paper. For details please see appendix A.2.
>
> | Method                            | Herpes              | Adeno                 | Noro               | Papilloma                    | Rota               |
> |-----------------------------------|------------------------|------------------------|------------------------|------------------------|------------------------|
> | $\mathrm{Ours (Opt)}$ ResNet      | **86.98** $\pm$ 1.92   | **47.85** $\pm$ 11.82  | **54.65** $\pm$ 4.94   | **70.02** $\pm$ 2.85   | **71.73** $\pm$ 3.51   |
> | $\mathrm{Ours (Opt)}$ ViT         | 48.66 $\pm$ 07.44      | 07.46 $\pm$ 05.21      | 10.44 $\pm$ 09.06      | 05.67 $\pm$ 08.05      | 04.50 $\pm$ 3.51       |
>
>
> As presented in the table above, the results show that the ViT especially underperforms for the very small negative stain data sets of the Adeno, Noro, Papilloma and Rota virus.

---

> ### Author Response · Authors · 2023-11-22
> **DINO-ViT in CutLer**
>
> In our experiments, we did not train a DINO-ViT for CutLer on EM images.
> In our experience with training models on EM images, we found that most often fine-tuning ImageNet pre-trained weights can lead to superior performance compared to fine-tuning models based on EM pre-trained weights [8].
> This can be due to the natural fit of ViTs to EM images (as you've mentioned) but also to the usually much bigger size of the pretraining dataset (exemplified by ImageNet with 1.2 million training images as opposed to CEM500k [8] with 500,000 training images).
> Fine-tuning the DINO-ViT on EM images is, therefore, a non-trivial task, warranting a more thorough investigation. The intricacies involved in achieving optimal performance in this context necessitate a comprehensive exploration of the challenges and potential adaptations for ViTs in the domain of EM image fine-tuning.

---

> ### Author Response · Authors · 2023-11-22
> **Label Distribution**
>
> We agree that the distribution of the training data is difficult to understand, especially given the evaluation using different data set sizes based on annotation times. We hence include a more informative table in appendix D.

---

> ### Author Response · Authors · 2023-11-22
> **References**
>
> [1] Xu, J., Hou, J., Zhang, Y., Feng, R., Zhao, R.W., Zhang, T., Lu, X. and Gao, S., 2022. Cream: Weakly supervised object localization via class re-activation mapping. In Proceedings of the IEEE/CVF Conference on Computer Vision and Pattern Recognition (pp. 9437-9446).
>
> [2] Wei, J., Wang, S., Zhou, S.K., Cui, S. and Li, Z., 2022, October. Weakly supervised object localization through inter-class feature similarity and intra-class appearance consistency. In European Conference on Computer Vision (pp. 195-210). Cham: Springer Nature Switzerland.
>
> [3] Weizeng Lu, Xi Jia, Weicheng Xie, Linlin Shen, Yicong Zhou, and Jinming Duan. Geometry constrained weakly supervised object localization. In European Conference on Computer Vision (ECCV), 2020
>
> [4] Jiang, Peng-Tao, et al. "Layercam: Exploring hierarchical class activation maps for localization." IEEE Transactions on Image Processing 30 (2021): 5875-5888.
>
> [5] Gao, Wei, et al. "Ts-cam: Token semantic coupled attention map for weakly supervised object localization." Proceedings of the IEEE/CVF International Conference on Computer Vision. 2021.
>
> [6] Su, Hui, et al. "Re-attention transformer for weakly supervised object localization." arXiv preprint arXiv:2208.01838 (2022).
>
> [7] Selvaraju, Ramprasaath R., et al. "Grad-cam: Visual explanations from deep networks via gradient-based localization." Proceedings of the IEEE international conference on computer vision. 2017.
>
> [8] Conrad, Ryan, and Kedar Narayan. "CEM500K, a large-scale heterogeneous unlabeled cellular electron microscopy image dataset for deep learning." Elife 10 (2021): e65894.

---

### Official Review · Reviewer_xKLD · 2023-11-06

**Soundness:** 3 good
**Presentation:** 3 good
**Contribution:** 3 good
**Rating:** 6
**Confidence:** 3

**Summary:**

This paper presents a domain-specific weakly supervised object detection method that relies on image-level annotations instead of bounding boxes. They use a pre-trained model to generate pseudo-labels for training, showing that these labels outperform other weak labeling methods and even ground truth labels in time-constrained scenarios.

**Strengths:**

- the paper is well-written and easily to follow.
- it proposed an relevantly simple but effective method for an impactful task. In their experimenst, aurthors sucessfully demonstrated the supriority over the consider baselines, includig supervised method as well as zero-shot learning with large scale pretrained models.
- the authors utilized the spatial information and explored an novel way to refine the localization neural networks provide.

**Weaknesses:**

The proposed method has potential to work for not only electron microscope images but other medical images. It will be interesting and also brings broader impact if authors can provide discussions around this.

**Questions:**

The current setup with Gaussian as a prior assumes that the object to detect is in a round shape. How easily it can be extended to different objects and how accurately it will work?

---

> ### Author Response · Authors · 2023-11-22
>
> Dear reviewer xKLD,
> Thank you for your valuable comments. We would like to address your comments one by one, as seen below. We are looking forward to your feedback.
>
> ## Broader impact
> We agree, that the proposed method has potential to work not only for the detection of virus particles in EM but other medical images, or other standard CV datasets.
> However, our approach is strategically crafted for the detection of particles in EM images, rather than aiming for a broad range of applications and object types. In our expanded evaluation, we demonstrate the efficacy of our method by surpassing the performance of more intricate state-of-the-art methods, even when incorporating virus size information into their frameworks.
> The observed superiority of our method can be attributed to the unique challenges posed by virus detection in EM, which diverge from the characteristics addressed by current state-of-the-art methods. Specifically:
>
> 1. Limited Annotated Data: Current state of the art approaches highly benefit from large data set sizes. These large amounts of annotated data are not available for virus detection in EM.
> 2. Non-Object-Centric Nature of EM Images: Current state of the art approaches are usually designed for more object-centered approaches. This does not apply to EM images where multiple instances of the same virus occur in a single micrograph.
> 3. Low Signal-to-Noise Ratio (SNR) in EM: Current state of the art approaches usually do not need to deal with low SNR images, as it is the case for EM.
>
> These observations underscore the significance of developing methods explicitly designed to excel in the specialized task of virus detection in electron microscopy. Our approach addresses these unique challenges, positioning it as a valuable contribution to the field by outperforming existing methodologies specifically tailored for this demanding context. We will include a more in depth discussion about this in the updated version of our paper in relation to our extended evaluation (see section 4.4)
>
>
> ## Round shape as a prior and discussion on braoder impact
> We believe that the extension of the proposed method to a wider range of applications is possible. In [1] the authors use a generator to predict ellipses which are then used to mask the input image for training of a detector with image-level labels only. Substituting the proposed circle by ellipses will open the possibility to detect objects with different aspect ratios. They use an additional size penalty as a loss to additionally optimize for the size.
> Combining the proposed method with [1] will most likely achieve a broader application of the approach.
> However, as discussed above, our evaluation shows the need for more specific approaches in the field of virus detection in EM. We hence leave the extension of the approach to a wider range of applications for future work.
>
>
> [1] Lu, Weizeng, et al. "Geometry constrained weakly supervised object localization." Computer Vision–ECCV 2020: 16th European Conference, Glasgow, UK, August 23–28, 2020, Proceedings, Part XXVI 16. Springer International Publishing, 2020.

---

### Author Response · Authors · 2023-11-22
**Overview**

Dear Reviewers,

We are grateful for your insightful feedback and constructive comments on our work. We are encouraged by the recognition of the uniqueness and simplicity of our proposed method. The emphasis on an intuitive heuristic adds a layer of accessibility to our technique. We believe that the simplicity of our approach does not compromise its effectiveness (as shown in our evaluation), making it potentially interesting to a wider audience.

Nevertheless, we concur with the identified shortcomings acknowledged in the review process. We would like to highlight the following adaptions of our work based on your valuable comments:

- Extension of the evaluation to a variety of other weakly supervised approaches
- Inclusion of the virus size into existing methods for a fair comparison
- Additional exploration of model architectures (particularly ViTs)
- Inclusion of discussion on low SNR and specific challenges inherent to EM
- Emphasis on the critical importance of the known virus size as a prerequisite for the proposed method's efficacy by the inclusion of an additional ablation study

Please note that we highlighted all changes in the updated pdf with a green background. Based on the extension of the evaluation and the more in-depth discussion, we needed to move ablations of the method into the appendix.

---

### Meta-Review · Area_Chair_eyae · 2023-12-11

**Metareview:**

The paper introduces a domain-specific weakly supervised object detection method for electron microscopy (EM) images, leveraging image-level annotations and pre-trained models for virus presence/absence prediction. The proposed approach generates pseudo-labels, reducing annotation costs and facilitating effective model training. Utilizing a shrinking receptive field in the optimization process enables direct virus particle extraction, independent of specific network architectures. The method's strengths, as highlighted in the reviews, include its clear presentation, structural coherence, and simplicity. It introduces an intuitive heuristic, demonstrating efficacy in virus detection in EM images through pre-trained models and iterative refinement. Ablation studies validate its superiority over other annotation-based methods, even outperforming ground truth labels in time-constrained scenarios. The paper's unique methodology and simplicity position it as a potentially impactful contribution to weakly supervised object detection in the context of EM images. The weaknesses identified in the reviews center on the narrow focus and limited exploration of the proposed weakly supervised object detection (WSOD) method. Concerns arise about its specificity to electron microscopy (EM) images and the lack of comparisons with other state-of-the-art WSOD techniques, potentially limiting its broader applicability. Reviewers call for exploration of alternative models, such as Vision Transformers (ViTs), and emphasize the need for a more in-depth discussion on the impact of low Signal-to-Noise Ratio (SNR) in EM images. Additionally, questions are raised about potential computational costs, especially in scenarios with numerous objects. During the rebuttal, the authors have well addressed these concerns. Therefore, AC recommends accept.

**Justification For Why Not Higher Score:**

Concerns revolve around the narrow scope, limited exploration, and specificity to EM images, with recommendations for broader comparisons and deeper discussions on challenges associated with different imaging modalities.

**Justification For Why Not Lower Score:**

The paper ended up getting 5,6,6,6 (two reviewers expressed an increase in the rating in their message to AC but did not update timely in the system yet). The proposed weakly supervised object detection (WSOD) method is consistently praised across reviews for its clarity, well-structured presentation, and simplicity. The manuscript's unique methodology and the simplicity of the proposed technique make it a potentially impactful contribution to the domain of WSOD, particularly in the context of EM images.

---

### Decision · Program_Chairs · 2024-01-16

Accept (poster)